# Spatial information allows inference of the prevalence of direct cell–to–cell viral infection

**Thomas Williams** [1], **James M. McCaw** [1,2], **James M. Osborne** [1] *

1 School of Mathematics and Statistics, University of Melbourne, Melbourne, Australia, 2 Centre for Epidemiology and Biostatistics, Melbourne School of Population and Global Health, University of Melbourne, Melbourne, Australia

* jmosborne@unimelb.edu.au

**Data Availability Statement:** Our code is freely available at https://github.com/thomaswilliams23/dual_spread_viral_dynamics_fitting.

## Abstract

The role of direct cell–to–cell spread in viral infections—where virions spread between host and susceptible cells without needing to be secreted into the extracellular environment—has come to be understood as essential to the dynamics of medically significant viruses like hepatitis C and influenza. Recent work in both the experimental and mathematical modelling literature has attempted to quantify the prevalence of cell–to–cell infection compared to the conventional free virus route using a variety of methods and experimental data. However, estimates are subject to significant uncertainty and moreover rely on data collected by inhibiting one mode of infection by either chemical or physical factors, which may influence the other mode of infection to an extent which is difficult to quantify. In this work, we conduct a simulation–estimation study to probe the practical identifiability of the proportion of cell–to–cell infection, using two standard mathematical models and synthetic data that would likely be realistic to obtain in the laboratory. We show that this quantity cannot be estimated using non–spatial data alone, and that the collection of data which describes the spatial structure of the infection is necessary to infer the proportion of cell–to–cell infection. Our results provide guidance for the design of relevant experiments and mathematical tools for accurately inferring the prevalence of cell–to–cell infection in *in vitro* and *in vivo* contexts.

## Author summary

Viruses are known to spread between host cells either via infection with cell–free virions or through direct cell–to–cell infection. The prevalence of cell–to–cell infection for different virus species is not well known, yet is of huge importance to therapeutic applications due to its resilience to drug interventions and the immune response. In this work, we investigated whether the proportion of infections from each mode of spread could theoretically be inferred from data using two standard mathematical models of viral dynamics with both modes of infection. By generating synthetic observational data and refitting using the models, we found that the proportion of cell–to–cell infections could not be obtained using models or data which did not account for the spatial structure of the infection. However, using a spatially–explicit model and (practically obtainable) observational data which measured spatial features of the infection, the proportion of infections from

**Funding:** TW's research is supported by an Australian Government Research Training Program (RTP) scholarship. JMM's research is supported by the Australian Research Council (DP210101920). JMO's research is supported by the Australian Research Council (DP230100380, FT230100352). The funders had no role in study design, data collection and analysis, decision to publish, or preparation of the manuscript.

**Competing interests:** The authors have declared that no competing interests exist.

the cell–to–cell route could be reliably inferred, even when collecting data from only small samples of the model tissue. This work will hopefully inform the development of experimental procedures and mathematical models to improve estimates of the prevalence of cell–to–cell infection.

## Introduction

Classically, viral infections have been assumed to spread among host cells through a process of viral secretion, diffusion, and reabsorption via the extracellular environment [1, 2]. In reality, however, a huge variety of the most medically important viruses—including influenza A, herpesviruses, hepatitis C, HIV and SARS–CoV–2—have all been observed to also spread between host cells using direct cell–to–cell mechanisms [3–5]. This mode of infection, which is mechanistically distinct from the conventional cell–free route, permits viruses or viral proteins to be trafficked directly between adjacent cells without ever leaving the cell membrane [4]. This is significant for multiple reasons. For one, the direct cell–to–cell route of infection is orders of magnitude more efficient than the cell–free route [6–8], and moreover is far better protected from immune or drug defences [5, 6, 9]. Cell–to–cell infection is considered one of the essential strategies of chronic viral infections like hepatitis C and HIV, and elevated cell–to–cell spread has been associated with increased pathogenicity in influenza and SARS-CoV-2 infections [6, 10]. Estimating the prevalence of cell–to–cell infection in different viral species is therefore of profound importance in therapeutic applications.

Over the last decade, a substantial quantity of experimental and modelling studies have attempted to quantify the relative contributions of the cell–to–cell and cell–free mechanism in infection with different viral species. Among these works, the most developed body of literature concerns HIV. Interdisciplinary studies led by Komarova [11] and Iwami [12] suggested that cell–to–cell and cell–free infections contribute roughly equally in HIV infection *in vitro*; more recent work by Kreger and colleagues [13], which also modelled the latent stage of infection in HIV, inferred a significantly higher rate of cell–free infection. In hepatitis C, modelling efforts led by Graw and Durso–Cain suggested that cell–free infection events were rare, yet worked synergistically with the cell–to–cell infection strategy to rapidly accelerate the overall rate of infection spread [7, 14]. Blahut and coworkers used modelling to quantify the proportion of the two modes of spread using *in vitro* experimental data, and claimed that as little as 1% of the infection events observed were due to to–cell infection [15]. Experimental work by Kongsomros and colleagues suggested that the proportion of cell–to–cell infections in influenza was low, but elevated in more pathogenic strains of the virus [6]. Experimental work in SARS-CoV-2 by Zeng and collaborators claimed that cell–to–cell infection represented around 90% of infections [10].

These estimates in the literature for the relative contribution of cell–to–cell spread in infection are subject to substantial uncertainty, and share the common limitation that they rely on experiments which block one of the modes of viral spread, compared to a control case where both routes of infection are active [6, 10–13, 15]. This inhibition can be implemented in a number of ways, such as by conducting infection assays in the presence of an antiviral agent or a physical barrier to viral diffusion like methylcellulose to block cell–free infection [6, 10], or by constantly shaking the cell culture to prevent the formation of virological synapses which enable cell–to–cell infection, in the case of HIV [11–13]. These approaches, however, share some common limitations. For one, the two modes of viral spread are known to interact synergistically, and the inhibition of one of the infection mechanisms invariably influences the

strength of the other mode of infection [8, 11, 12]. For instance, in the case of the static and shaking assays for HIV infection described above, Komarova and colleagues estimated that shaking the cell culture increased the rate of cell–free infection by around 1.33 times [11]. A second shortcoming of this approach is its inapplicability to *in vivo* settings. In living organisms, host toxicity or simple practicality prevents the use of most interventions to block one mode of viral spread, such as treating cells with methylcellulose or continuously shaking the cell population, yet the relative contribution of the two modes of infection may be substantially different *in vivo* compared to in cell culture. For instance, Dixit and Perelson estimated that in human hosts, roughly 90% of HIV infection was due to cell–to–cell spread [16], whereas estimates from *in vitro* data placed this figure at around 50% [11–13].

Only a few studies have attempted to infer the balance of the two modes of infection spread from data where both mechanisms are unimpeded. Imle and collaborators studied HIV infection in cell cultures embedded either in suspension or a 3D collagen scaffold, and calibrated an ODE model to the data to attempt to infer the relative contribution of the two modes of infection from virion and cell count data [17]. The authors suggested that the inference implied almost all infection in the suspension was due to cell–free infection, however, the 95% confidence interval for the proportion of cell–free infection encompassed virtually the whole range from 0 to 100% [17]. In hepatitis C, Kumberger and colleagues demonstrated modifications that can be made to a standard ordinary differential equation (ODE) model of viral dynamics in order to better describe cell–to–cell infection, but were nonetheless unable to satisfactorily infer the prevalence of cell–to–cell infection from synthetic data where the two modes of infection occurred simultaneously [18]. The authors moreover did not examine whether estimates of this quantity were improved or weakened when the true balance of the two mechanisms in the synthetic data was changed [18]. The limits of identifiability of the proportion of cell–to–cell infection—under different conditions, using different models, and based on different sources of observational data—has not been systematically studied.

Here, we conduct simulation–estimation studies using two mathematical models for viral infections with two modes of spread: one non–spatial ODE system and one spatially–explicit multicellular model. In both cases, we generate synthetic data using the model in combination with an observational model, and attempt to re–estimate the prevalence of cell–to–cell infection from the resulting observations. We repeat this process under a range of conditions and with different types of available data for fitting. Our results provide an important background for the practical identifiability of the cell–to–cell infection prevalence, and offer guidance for the design of models and experimental systems best equipped to learn this quantity. It is important to mention that the analysis which we conduct here is limited to infections of static tissues, and does not extent to infections in motile cell populations, such as HIV. Since, in this case, the migration of target cells enables well–mixed conditions, there is no notable spatial structure to infection and thus the resulting dynamics are less easily distinguished.

In this work we take particular inspiration from the work of Kongsomros and colleagues [6]. In their work, the authors conduct a series of experiments where "donor" cells infected with influenza are added to a well of "recipient" cells, labelled with a membrane dye, and infection allowed to spread under a given set of experimental conditions. At various times, wells are harvested and fixed, then stained with fluorescent anti viral–NP antibody to identify the infected recipient cell population. In the present work, we will take the fluorescent cell proportion, following the construction given here, as our primary source of observational data. We provide further discussion of our choice of data source in Discussion.

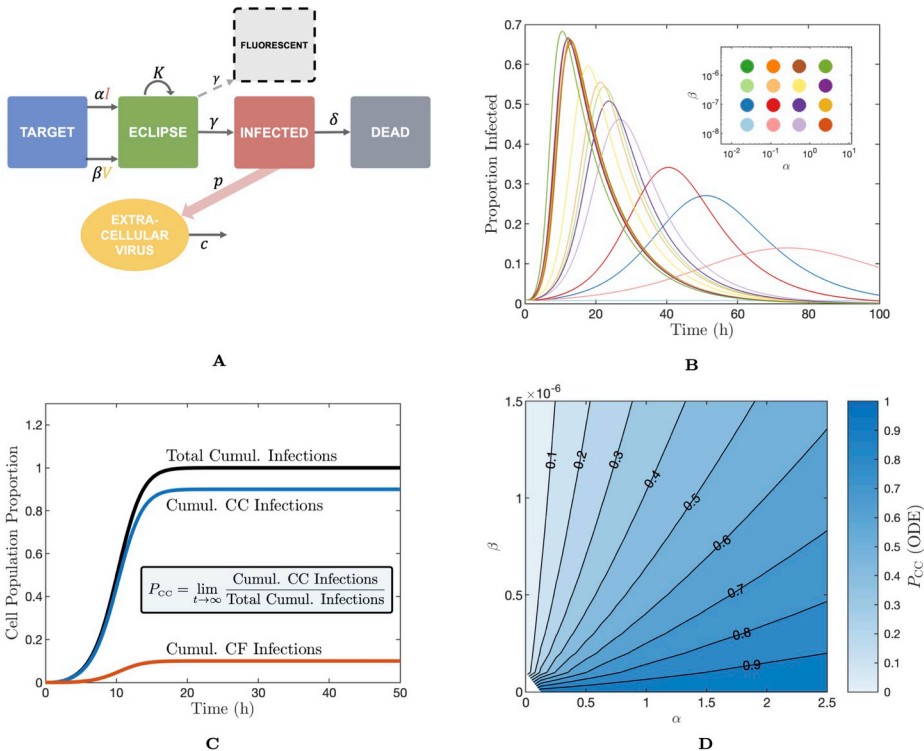

**Fig 1. Schematics for the ODE model and proportion of infections from the cell–to–cell route.** (A) Schematic of the ODE model. (B) Proportion of infected cells over time as predicted by the ODE model for an array of values of $\alpha$ and $\beta$ between zero and 2.5 and $2 \times 10^{-6}$, respectively. The parameter values sampled to generate the plot are shown in the inset. (C) Calculation of $P_{CC}$. We keep track of the proportion of the cell population which has been infected by the cell–to–cell (CC) and cell–free (CF) infection over the course of infection. We define $P_{CC}$ as the proportion of infections arising from the CC route at long time. (D) $P_{CC}$ contour map on $\alpha$–$\beta$ space for the dual–spread ODE model. $\alpha$ and $\beta$ have units of $h^{-1}$ and $(TCID_{50}/ml)^{-1}h^{-1}$, respectively.

## Results

### In the presence of observational noise, the prevalence of cell–to–cell infection spread cannot be determined from fluorescence time series data alone

We sought to investigate whether an ODE model incorporating both cell–free viral infection and cell–to–cell infection, could be used to infer the balance of the two modes of spread, given a time series of observations of the fluorescent proportion of the cell population as in Kong-somros *et al.* [6]. We exhibit the basic properties of the ODE model in Fig 1 (the model is fully described in Methods "An ODE model for dual–spread dynamics"). Fig 1A shows the basic structure of the model and the parameters governing the model. We apply a standard target cell–limited model framework with a latent compartment and two modes of infection. That is, initially susceptible cells may become infected either through cell–to–cell infection—at a rate proportional to the infected proportion of the cell population—or through infection by cell–free virus—at a rate proportional to the quantity of extracellular virus in the system. Once initially infected, cells enter the first of $K$ eclipse sub-stages (such that the duration of the eclipse stage is gamma–distributed, instead of exponentially–distributed, see [19, 20]), before becoming productively infected, at which stage they begin producing extracellular virus. Productively infected cells then die. We assume that cells become detectably fluorescent once they become

productively infected, but that they remain fluorescent after death over the time scale of simulations, as observed in Kongsomros *et al.* [6].

Throughout this work we will take the majority of the model parameters to be fixed (which is discussed in Methods "An ODE model for dual–spread dynamics"), aside from the two parameters governing the rates of cell–to–cell and cell–free infection, $\alpha$ and $\beta$, respectively. Fig 1B shows the dynamics of the infected cell proportion over time using the ODE model with a range of $\alpha$ and $\beta$ values (throughout this work, $\alpha$ and $\beta$ have units of $h^{-1}$ and $(TCID_{50}/ml)^{-1}h^{-1}$, respectively). We can quantify and describe the overall rate of infection progression by the exponential growth rate $r$ (units of $h^{-1}$). This quantity, well established in the theory of both between–host and within–host infection dynamics, describes the initial rate of exponential expansion of the infected (or fluorescent) population [21, 22]. For further details refer to Methods "Exponential growth rate—$r$".

We applied simulation–estimation techniques to investigate whether $\alpha$ and $\beta$ could be inferred from the fluorescent cell time series of the model. We first selected three sets of $(\alpha, \beta)$ pairs resulting in different proportions of infections arising from each mechanism. Specifically, if we label the final fraction of infections arising from the cell–to–cell route as $P_{CC}$, we construct lookup tables on $\alpha$–$\beta$ space for this quantity, and use this to compute $(\alpha, \beta)$ pairs corresponding to $P_{CC}$ values of approximately 0.1, 0.5, and 0.9, with a fixed exponential growth rate $r$ of 0.52 in each case to ensure the overall dynamics progressed at a comparable rate. We show a graphic of the computation of $P_{CC}$ in Fig 1C, and a contour map on $\alpha$–$\beta$ space for the ODE model in Fig 1D. For further details on $P_{CC}$, refer to Methods "Proportion of infections from the cell–to–cell route—$P_{CC}$".

For each of the specified values of $(\alpha, \beta)$, we simulated the ODE model and, following Kongsomros and colleagues, we computed the fluorescent cell proportion $F(t)$—that is, the cumulative proportion of the initially susceptible population that has become infected—at $\mathbf{t} = \{3, 6, 9, \ldots, 30\}$h [6]. We then applied an observational model to this data to simulate the experimental process, by assuming a cell population size $N_{sample}$, and overdispersed noise modelled by a negative binomial distribution. We take $N_{sample} = 2 \times 10^5$ as in Kongsomros *et al.* [6] and set the dispersion parameter $\phi = 10^2$, selected to impose a modest amount of noise on our observations, leading to the observed data vector $\mathcal{D}$. We specify the observation model in full in Methods "Simulation–estimation", and explore the role of observational noise in more detail in S1 Text and S1 Fig.

Having obtained our observed data $\mathcal{D}$, we run a No U–Turn Sampling (NUTS) Markov Chain Monte Carlo (MCMC) algorithm [23] to obtain posterior density estimates for $\alpha$ and $\beta$. For each $(\alpha, \beta)$ pair to estimate, we run ten replicates of the simulation–estimation process. That is, for each replicate we apply random observational noise to the true fluorescence data and then re–estimate $\alpha$ and $\beta$ using four independent and randomly seeded chains. We draw 2000 samples from each chain and discard the first 200 samples as a burn–in. We assume uniform priors for $\alpha$ and $\beta$ on $[0, 2.5]h^{-1}$ and $[0, 2 \times 10^{-6}](TCID_{50}/ml)^{-1}h^{-1}$respectively, and assume a negative binomial likelihood. Further details of this simulation–estimation process are specified in Methods "Simulation–estimation".

In Fig 2, we show the results of this fitting process. In Fig 2A–2C, we show heat maps of the density of posterior samples in $(\alpha, \beta)$ space from each chain of a single replicate fit. We do so for each of the three target parameter pairs. As a visual aid, we also plot the $(\alpha, \beta)$ contours corresponding to the true $P_{CC}$ value and the true $r$ value in each case. These plots show that the posterior samples for each pair of target parameters are spread out along the true $r$ contour. While some samples are close to the target parameter pair, the chains do not appear to converge at this point. In S3 Fig, we show an equivalent plot to Fig 2A–2C as a scatter plot of accepted samples, which confirms that the chains are indeed well–mixed. In Fig 2D, for each

FITTING FLUORESCENCE DATA – ODE MODEL

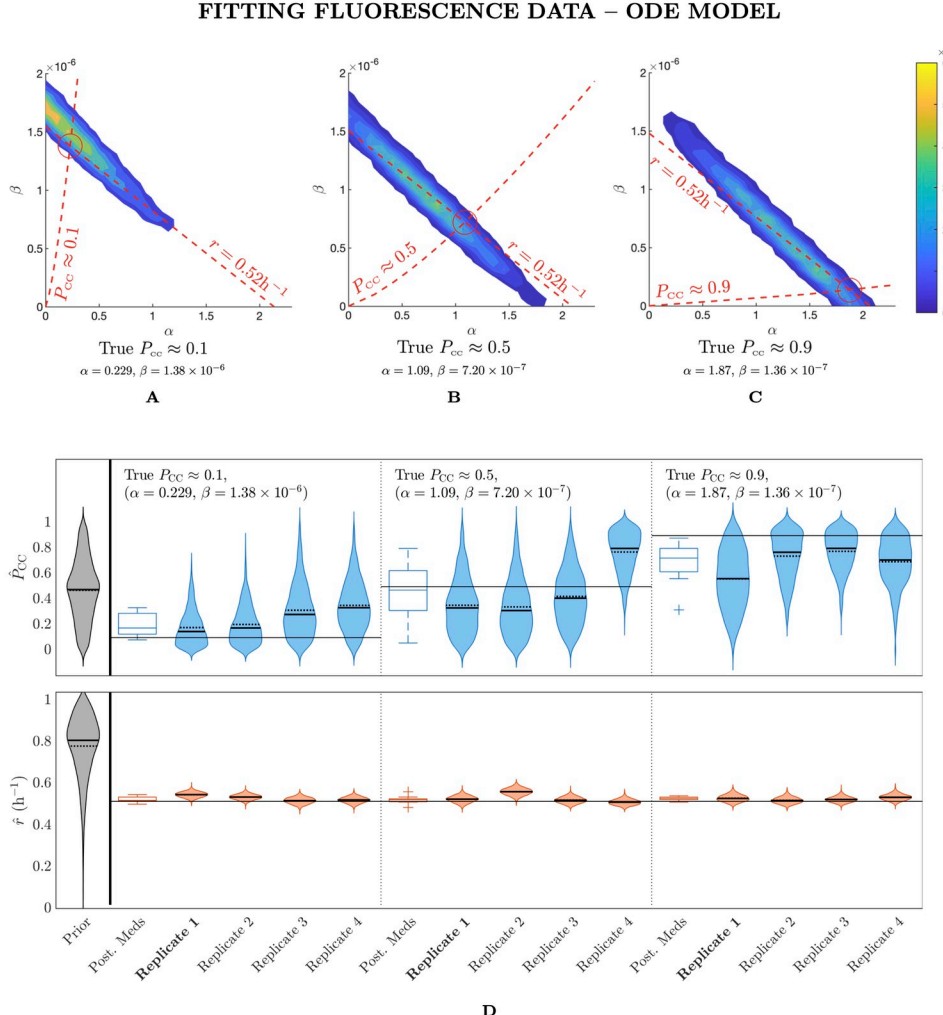

**Fig 2. Fitting fluorescence data with the ODE model does not permit inference of the prevalence of cell–to–cell spread.** (A)–(C) Posterior density as a contour plot in $\alpha$–$\beta$ space for a fit to fluorescence data with the ODE model where the true $P_{CC} \approx 0.1, 0.5, 0.9$ and the value of $r$ is held fixed. Density is shown for the 1800 samples from each chain after burn–in for a single replicate. We only show densities above a threshold value of $10^{-4}$. (D) Prior density and posterior densities from individual replicates for $r$ and $P_{CC}$ both with typical observational noise. We repeat this for three sets of parameters resulting in $P_{CC}$ values of 0.1, 0.5 and 0.9 with a fixed $r$ value. Dashed and solid horizontal lines mark the mean and median values respectively. We also show a box plot of the distribution of posterior medians across all replicates. There are ten replicates in total at each value of $P_{CC}$, of which we display four. The marginal posterior densities of $\alpha$ and $\beta$ are shown in S4 Fig. $\alpha$ and $\beta$ have units of h$^{-1}$ and (TCID$_{50}$/ml)$^{-1}$h$^{-1}$, respectively.

target parameter pair, we show violin plots of the posterior distributions of $P_{CC}$ and $r$ for four replicate fits, along with a box plot of the posterior medians across all ten replicates. We also show the prior density of both of these quantities in grey. Fig 2D shows that while $r$ is well esti-mated compared to its prior distribution—regardless of the choice of target parameters—$P_{CC}$ cannot be practically identified even when a conservative amount of observational noise is present. While, at least for the case where the target $P_{CC} = 0.1$, the distribution of posterior medians can be somewhat accurate, posterior distributions from individual replicates are fre-quently far from the true value. Importantly, some of these posterior distributions have a high degree of precision, yet are inaccurate, for instance, Replicate 4 for the case where the target

$P_{CC}$ = 0.5. The individual posterior distributions for $\alpha$ and $\beta$, which we show in S4 Fig, show a similar practical unidentifiability. While the mode of the distributions roughly follows the true values of these parameters, the sample densities are dispersed widely, hence confidence intervals on $\alpha$ and $\beta$ are wide. Overall, this experiment indicates that, when even a modest degree of observational noise is applied to the fluorescence data, only the exponential growth rate $r$ can be accurately estimated: the proportion of infections arising from each mode of spread is lost in the observational process.

We investigated the role of the level of observational noise in determining the quality of estimates of $P_{CC}$ and $r$ using the ODE model (for full details, see S1 Text). We found that for higher values of the dispersion parameter $\phi$ than we show here (that is, with less observational noise), estimates of $P_{CC}$ were overall closer to the true value, however, the distribution of estimate medians still showed not insignificant variance, even when virtually all observational noise was removed. Subject to a higher level of observational noise, estimates of $P_{CC}$ were almost entirely random. We show these results in full in S1 Fig.

## Using a spatial model with spatial data, the balance of the modes of infection spread can be accurately inferred

We sought to apply a similar simulation–estimation procedure to a spatially–structured model of infection, to investigate whether a model capable of describing the actual structure of infection would provide better estimates of the proportion of each infection mechanism. We constructed an agent–based spatial model with an equivalent structure to the ODE model used in the previous result, where transitions between compartments of the model are replaced by probabilities of discrete cells, occupying specific positions in space, changing between states analogous to those in the ODE model. The notable difference in this construction is that while we still model cell–free infection based on a *global* extracellular viral reservoir, we now model cell–to–cell infection as a spatially *local* process. Specifically, we assume that the probability of cell–to–cell infection of a given cell is based on the infected proportion of its neighbours, instead of the global infected cell population as in the ODE model. This reflects the assumption—based on current biological understanding—that cell–free virions spread rapidly over the size of tissue we seek to model, whereas cell–to–cell infection is possible only between adjacent cells [3]. This process is illustrated in Fig 3A. Fig 3A shows a schematic of the spatial model, and illustrates the alternate formulation of the cell–to–cell infection mode. Note that, as illustrated in the schematic, cells are packed in a hexagonal lattice, which reflects the biological reality of epithelial monolayers and moreover ensures that adjacency between cells is well–defined. Full details of the spatial model can be found in Methods "A multicellular spatial model for dual–spread dynamics".

In addition to the fluorescent proportion metric we introduced in the previous result, we developed an additional metric for the spatial model to describe the extent to which infected cells were clustered together. This metric, which we term $\kappa(t)$, describes the mean proportion of neighbours of the fluorescent cells which are also fluorescent at time $t$. In Fig 3C we show a schematic which illustrates the computation of the fluorescent neighbour fraction at a number of fluorescent cells in a cell sheet. We define $\kappa(t)$ explicitly in Methods "Clustering metric— $\kappa(t)$". $\kappa(t)$ has the property that when it is large, fluorescent cells tend to be clustered together and the infection is highly localised, whereas if it is small, the infection is diffuse.

In Fig 3D–3G, we demonstrate the behaviour of the spatial model under three $(\alpha, \beta)$ parameter pairs, chosen to result in a $P_{CC}$ of approximately 0.1, 0.5, and 0.9, and to reach a peak infected cell fraction at approximately 18h. In Fig 3D, we visualise a section of the cell grid at a series of time points. We do so by assigning a unique index $j = \{1, 2, \ldots, N_{init}\}$ to each of the

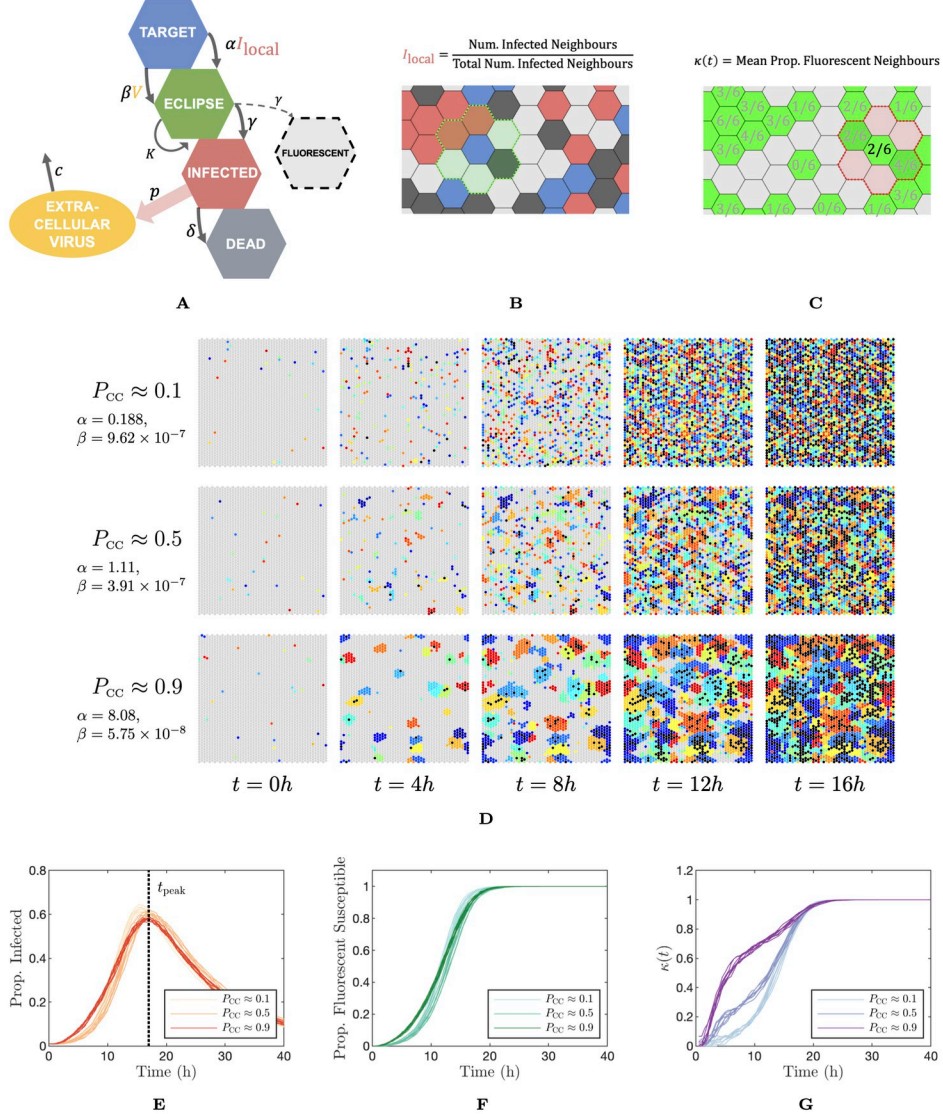

**Fig 3. Dynamics and metrics of the spatial model.** (A) Schematic of the spatial model. The model follows the same structure as the ODE model with the exception that cell–to–cell infection is based on the proportion of a cells neighbours which are infected. (B) Cartoon of the calculation of infected neighbour proportion. (C) $\kappa(t)$ is our clustering metric, computed as the mean proportion of neighbours of the fluorescent cells which are also fluorescent. (D) Typical time evolution of the cell grid using the spatial model under three $\alpha$–$\beta$ combinations, resulting in $P_{CC}$ values of approximately 0.1, 0.5, and 0.9. Parameters were chosen such that the peak infected cell population is reached at approximately the same time in each instance. Initially infected cells are flagged with a unique colour and infections resulting from that lineage of cells are assigned the same colour. Target cells are marked in grey and dead cells in black. (E), (F), (G) Proportion of cell sheet infected, proportion of susceptible cells which are fluorescent over time, and the clustering metric $\kappa(t)$ respectively. We show eight simulations for each of the $\alpha$–$\beta$ parameter pairs described above. $\alpha$ and $\beta$ have units of $h^{-1}$ and $(TCID_{50}/ml)^{-1}h^{-1}$, respectively.

$N_{init}$ initially infected cells and the extracellular virus they produce. Then, every time a susceptible cell is marked for infection during a simulation, we compute the probability that it was caused by each of the $N_{init}$ viral lineages, and determine the lineage assigned to that cell. Infected cells are then coloured by their lineage. Once a cell dies, we change its colour to black. This construction allows us to visualise the spread of infection in space. Fig 3D shows that

when cell–to–cell infection dominates, infection plaques are tightly clustered and infected cells of the same lineage tend to be found closer together. When cell–free infection dominates, there is no particular structure to the colouring of the cell sheet. In Fig 3E–3G, we show time series for the spatial model under the same three parameter schemes as discussed above: the proportion of the cell population which is infected over time, the fluorescent cell curve as discussed in the previous section, and the clustering metric $\kappa(t)$. These time series indicate that even though the different parameter regime lead to vastly differently-structured infections—as can be seen in Fig 3D—their infected and fluorescent cell count dynamics as a time series are relatively similar, although there is some variation in the initial uptick of infection in the case where $P_{CC}$ is large. By contrast, the time series for $\kappa(t)$ shows substantial variation between the parameter values corresponding to low, roughly equal and high values of $P_{CC}$.

Since in the spatial model, cell–to–cell infection is constrained to act locally, infections that spread mainly through cell–to–cell infection are forced to spread radially. The size of the resulting infected cell population, therefore, grows in a non-exponential manner. For this reason, the exponential growth rate $r$ is not well–defined in the case of the spatial model. As an alternative metric of the rate of growth of the infected cell population, we simply use the time of the peak infected cell population, which we label as $t_{peak}$. Since this, like $P_{CC}$, cannot be well-estimated *a priori*, we again resort to computing a lookup table of mean $t_{peak}$ values on $\alpha$–$\beta$ space. For full details on the construction of these lookup tables and their corresponding surface plots, refer to Methods "Proportion of infections from the cell–to–cell route—$P_{CC}$".

We computed $(\alpha, \beta)$ pairs for the spatial model which result in $P_{CC}$ values of approximately 0.1, 0.5, and 0.9 and a common value of $t_{peak}$ of approximately 18h, analogous to the values selected for the ODE model in our previous fitting experiment. For each of these parameter pairs, we ran simulations of the spatial model and reported the fluorescent proportion of the susceptible cells as well as the clustering metric $\kappa(t)$ at times $\mathbf{t} = \{3, 6, 9, \ldots, 30\}$h, one time point per simulation. This model reflects the destructive experimental observation process. We provide full details of the observational model in Methods "Simulation–estimation". The resulting observations collectively form our observed data vectors $\mathcal{D}^{\text{spatial}}_{\text{fluoro}}$ and $\mathcal{D}^{\text{spatial}}_{\text{cluster}}$. We then used Population Monte Carlo (PMC) methods to re–estimate $\alpha$ and $\beta$ (full details in Methods "Simulation–estimation") given this synthetic observational data. For each of the three target $(\alpha, \beta)$ pairs, we ran four replicates of the data generation and fitting process.

We show the results of this experiment in Fig 4. Fig 4, which follows a similar layout to Fig 2, shows that with the addition of clustering metric data, $P_{CC}$ can now be robustly inferred using the spatial model. In Fig 4A–4C, we plot heat maps of the density of the final accepted posterior samples for $\alpha$ and $\beta$ in $\alpha$–$\beta$ space for the three target parameter pairs, resulting in $P_{CC} \approx 0.1, 0.5, 0.9$. These plots show posterior density distributed compactly around the true values of $(\alpha, \beta)$, instead of being dispersed along a $t_{peak}$ contour as in the previous simulation–estimation. In Fig 4D, we show the weighted posterior distributions of $P_{CC}$ and $t_{peak}$ for individual replicates along with the distribution of weighted posterior means across replicates. As before, $t_{peak}$ is still extremely well estimated in each case, however, now the posterior distributions for $P_{CC}$ are also very accurate to the true value. Moreover, the posterior distributions for individual replicates are concentrated on the true values of $P_{CC}$ with only modest confidence intervals, and the distributions of weighted mean estimates across replicates are extremely precise to the true values, meaning that carrying out inference with only a single data stream (as opposed to aggregating across multiple observations) was sufficient to estimate both $P_{CC}$ and $t_{peak}$. This was not the case with the ODE model. We also show the individual posterior distributions for $\alpha$ and $\beta$ in S5 Fig. S5 Fig shows a sharp peak of probability density around the true value of both $\alpha$ and $\beta$ for each value of $P_{CC}$, especially when that mode of infection is minimal.

FITTING FLUORESCENCE AND CLUSTERING DATA – SPATIAL MODEL

**Fig 4. Fitting fluorescence and clustering data with the spatial model allows the prevalence of cell–to–cell spread to be determined.** (A)–(C) Posterior density as a contour plot in $\alpha$–$\beta$ space for a fit to fluorescence and clustering data where the true $P_{CC} \approx 0.1$, 0.5, 0.9 and the infected cell peak time is held fixed at approximately 18h. We only show densities above a threshold value of $10^{-4}$. (D) Prior density and posterior densities from individual replicates for infected peak time ($t_{peak}$) and $P_{CC}$ with target parameters as specified in (A)–(C). Dashed and solid horizontal lines mark the weighted mean and median values respectively. We also show a box plot of the distribution of posterior weighted means across all four replicates in each case. The marginal posterior densities of $\alpha$ and $\beta$ are shown in S5 Fig. The replicates in bold are those plotted in (A)–(C). $\alpha$ and $\beta$ have units of h$^{-1}$ and (TCID$_{50}$/ml)$^{-1}$h$^{-1}$, respectively.

We note that estimates for $P_{CC}$ are especially sharp when the true value of $P_{CC}$ is higher, suggesting that the dynamics in this high cell–to–cell scheme are particularly distinguishable.

To test whether our results were dependent on the inclusion of the secondary data source, the clustering metric $\kappa(t)$, we performed another set of simulation–estimations using the same methods as above, this time using only the fluorescence data (full details in S7 Text). We show the results of this fitting experiment in S6 Fig. This figure shows that, without the use of the clustering metric, estimates for $P_{CC}$ are again very poor, while estimates for $t_{peak}$ remain reasonably precise. This result, which mirrors what we observed with the ODE model, suggests

that fluorescence data alone is not sufficient to imply the balance of the two modes of viral spread, even for the spatial model. We provide more discussion on this point in S4 Text.

The observational model used in our analysis here aims to recreate the noise incurred in an experimental setting. As such, we obtain our observational time series data by sampling one observation from each of a set of independent stochastic runs of the spatial model. This reflects the destruction of the cell culture in the observation process. However, it is certain that in experimental settings the observational process will incur additional noise than we have explicitly accounted for in this model. As such, we repeated the fitting process shown here after applying an additional negative binomial observational noise layer (the same as used for the ODE model) to both the fluorescence and clustering data. We discuss our results in S2 Text and S2 Fig. S2 Fig shows that estimates for both $P_{CC}$ and $t_{peak}$ with the spatial model are robust to a substantial degree of observational noise, especially when compared to applying the same levels of noise to the data and performing inference under the ODE model (shown in S1 Text). This finding suggests that estimates of $P_{CC}$ using this approach is resilient to additional noise which may be incurred in an experimental setting.

## The proportion of cell–to–cell spread can be inferred from diffusion–limited observational data within reasonable limits

So far, we have relied on the assumption that the diffusion of extracellular virions across the model tissue is sufficiently fast that the density of free virus in the system can be approximated as uniform. Clearly, this is a simplification of the biological reality. While the true value of the diffusion coefficient for free virions in media of differing properties is difficult to estimate [17, 24, 25], it is reasonable to assume that extracellular virions are to some extent constrained in the rate at which they spread across the tissue. At very slow diffusion, it may be that cell–free infection is indistinguishable from cell–to–cell infection. It is as yet unclear how well the approach we discuss here might apply to data collected from a diffusion–limited system.

To explore this, we developed an extended spatial model to include a spatially–structured viral density. We assume viral density is secreted continuously by infected cells uniformly in space across their surface, and free virus diffuses across the tissue according to linear diffusion with coefficient $D$. Throughout this work, we use units of $CD^2h^{-1}$ for $D$, where CD is a cell diameter, taken here to be approximately $10\mu$m for a typical respiratory epithelial cell [26]. For full details of the extended model, refer to Methods "A multicellular spatial model for dual–spread dynamics", and S5 Text for implementation.

We investigated the behaviour of the extended model for varying values of the diffusion coefficient, and the proportion of cell–to–cell infection. The time of the peak infected proportion was held fixed at 18h as in the previous result. Again, $\alpha$ and $\beta$ values corresponding to specified values of $P_{CC}$ and $t_{peak}$ were obtained using lookup tables, however, since these metrics are influenced by the choice of diffusion coefficient $D$, we constructed new lookup tables for each value of $D$ tested. In Fig 5A, we show a visualisation of the cell grid at the completion of infection for a range of values of the diffusion coefficient and the $P_{CC}$. Here, we follow a similar approach to Fig 3D, where we assign each initially infected cell a unique colour and colour each newly infected cell by the lineage that infected it. In Fig 5A, we colour each cell—including dead cells—by the lineage with which they were infected. Fig 5A shows that when diffusion is very small ($\mathcal{O}(10^{-1})CD^2h^{-1}$) the difference in the final grid state is almost imperceptible between different cell–to–cell infection fractions. In each case, the grid is divided into large, single–colour foci, indicating that cell–free infections under this scheme are all extremely close to the infecting cell. As the diffusion coefficient increases to around $\mathcal{O}(10^0) - \mathcal{O}(10^1)CD^2h^{-1}$, the edges of single–colour foci become frayed in low $P_{CC}$ cases and the grid structure is more

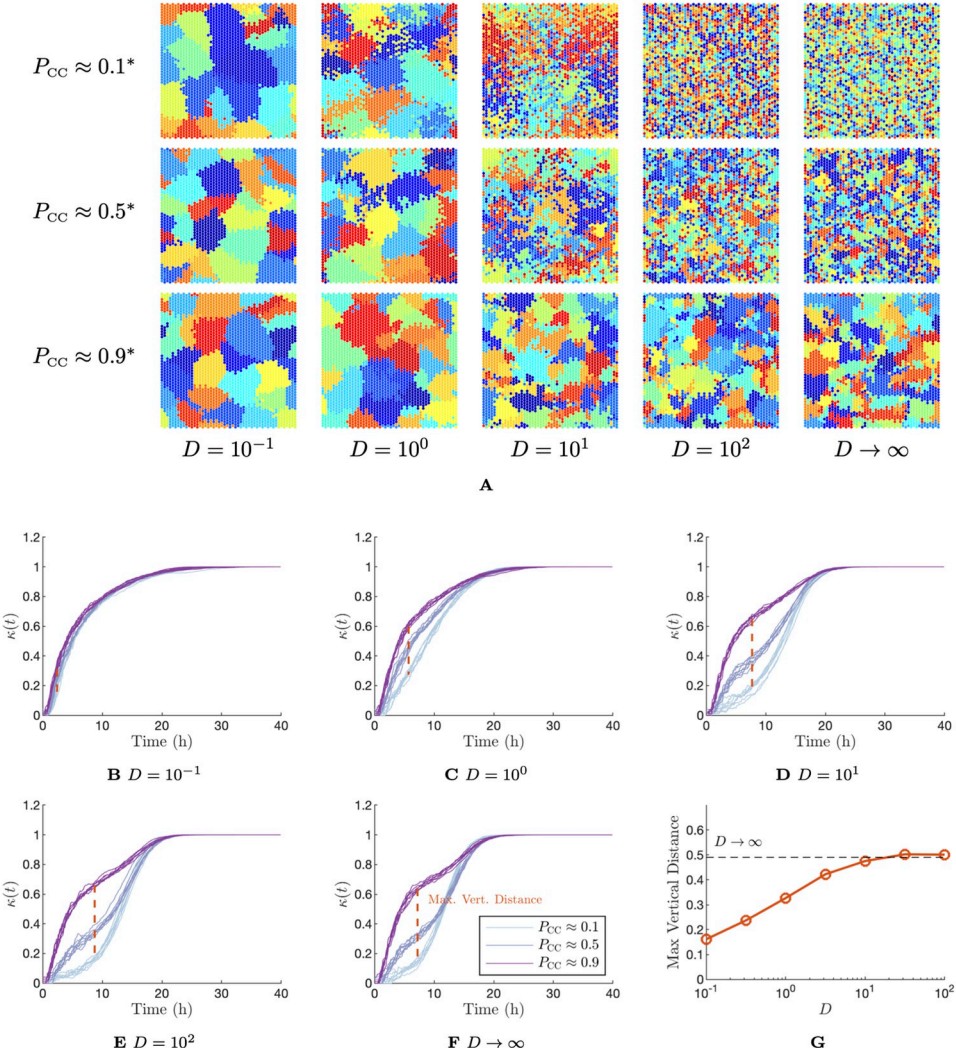

**Fig 5. Dynamics of the spatial model with diffusion–limited spread of extracellular virus.** (A) Final grid state following infection with the specified parameters. Initially infected cells are flagged with a unique colour and infections resulting from that lineage of cells are assigned the same colour. Here we show the final state of the cell grid, with each cell coloured by the lineage which infected it. *$\alpha$ and $\beta$ values computed from lookup table for relevant diffusion coefficient, ensuring a time of peak infected cell proportion at approximately 18h and the indicated proportion of cell–to–cell infection. $\alpha$ and $\beta$ values for each value of $D$, $t_{\text{peak}}$ and $P_{\text{CC}}$ used are specified in Table A in S5 Text. (B)–(F) The clustering metric, $\kappa(t)$ for the same diffusion coefficients and $\alpha$ and $\beta$ values as in (A). We show results from eight simulations in each case. (G) Maximum vertical distance between the mean $\kappa(t)$ curves for $P_{\text{CC}} = 0.9$ and $P_{\text{CC}} = 0.1$ for varying diffusion coefficients. $D$ has units of $\text{CD}^2\text{h}^{-1}$, where CD is a cell diameter.

distinct from the high $P_{\text{CC}}$ cases. For larger diffusion coefficients $\mathcal{O}(10^2)\text{CD}^2\text{h}^{-1}$, the grid states cannot be distinguished from that of the infinite diffusion (uniform virus) case.

In the previous result, we found that, while the time series for the proportion of infected cells in the sheet could not practically be distinguished for varying values of $P_{\text{CC}}$ (provided the infected peak time was held fixed), the corresponding time series for the clustering metric $\kappa(t)$ were clearly separated. Including this metric in our observational data therefore enabled the $P_{\text{CC}}$ to be inferred. As such, we computed $\kappa(t)$ time series for the same range of diffusion and $P_{\text{CC}}$ values as in Fig 5A to test if such a distinction would be preserved. We ran eight

simulations of the extended spatial model for each $D$–$P_{CC}$ combination, and show the resulting $\kappa(t)$ time series in Fig 5B–5F. Fig 5B–5F show that for diffusion coefficients of $D = 10CD^2h^{-1}$ and above, the wide variation between time series for varying $P_{CC}$ values is retained. Even for diffusion coefficients as low as $D = 1CD^2h^{-1}$, there is still a noticeable distinction between the curves, however, at $D = 0.1CD^2h^{-1}$ there is very little variation. We quantify the variation between the curves by computing the maximum vertical variation between the low and high $P_{CC}$ curves ($P_{CC} = 0.1$ and $P_{CC} = 0.9$, respectively) for each diffusion coefficient. We plot these in Fig 5G. Fig 5G confirms that for $D \geq 10CD^2h^{-1}$, there is as much distance between the curves as for the infinite diffusion case, but that this distance is lost rapidly for $D < 1CD^2h^{-1}$. These results suggest that it is reasonable to expect that $P_{CC}$ should be recoverable for a wide range of diffusion coefficients, including biologically likely values [24, 25].

We next carried out another round of simulation–estimations, where we generated diffusion–limited synthetic observational data using the extended spatial model under a range of values for the extracellular viral diffusion coefficient. We then use the (basic) spatial model—with diffusion misspecified as infinite—to re–fit the generated data. We computed target $(\alpha, \beta)$ parameter pairs corresponding to $P_{CC} = 0.1, 0.5, 0.9$ and $t_{peak} = 18h$ separately at each value of $D$ (these are specified in Table A in S5 Text). Moreover, since when diffusion is small the $(\alpha, \beta)$ pairs corresponding to this peak time exceed the support of the prior distributions for $\alpha$ and $\beta$ as defined for the previous results, we conduct this series of simulation–estimations using wider prior distributions. Specifically, we take $\alpha_{max} = 40h^{-1}$ and $\beta_{max} = 5 \times 10^{-5}(TCID_{50}/ml)^{-1}h^{-1}$, following the definition in Methods "Simulation–Estimation". Aside from these adjustments, these simulation–estimations were otherwise conducted using the same methods as in the previous result (summarised in Fig 4). We show the results of these simulation–estimations in Fig 6. Here we plot, as in previous figures, weighted posterior distributions for $P_{CC}$ for each value of the diffusion coefficient. For each of these values, we show the weighted posterior distributions for each replicate as well as a box plot of the weighted means across the replicates. As a reference, we also include our previously–discussed results for the case where the observational data is generated with uniform virus (infinite diffusion). We show the analogous plot for the time to the peak infected proportion, $t_{peak}$, in S8 Fig, which shows, as in previous results, that $t_{peak}$ is again well–estimated across each replicate, regardless of the value of the diffusion coefficient. By contrast, Fig 6 shows that the quality of estimation of $P_{CC}$ is highly dependent on the value of the diffusion coefficient. In general, the quality of estimates dramatically decreases for smaller diffusion coefficients. Results for $D \geq 10CD^2h^{-1}$ approach the quality of fit obtained for the infinite diffusion case, however, there is a radical departure from the true values of $P_{CC}$ for estimates where $D = 0.1CD^2h^{-1}$ or $1CD^2h^{-1}$.

While it is difficult to quantify the true value of the extracellular viral diffusion coefficient, Stokes–Einstein estimates for $D$ for influenza or SARS–CoV–2 virions in water at room temperature or plasma at body temperature have been computed to be approximately $216CD^2h^{-1}$ and $144CD^2h^{-1}$ respectively, assuming a cell diameter of approximately $10\mu m$ [25–27]. These diffusion coefficients are certainly sufficiently large to enable our approach here to apply, however, we note that several authors have assumed viral diffusion coefficients in various media to be orders of magnitude lower than these values (around $\mathcal{O}(1) - \mathcal{O}(10)CD^2h^{-1}$) [24, 25], in which case our approach may offer less precision in estimates of $P_{CC}$.

The $P_{CC} = 0.1$ case is estimated extremely poorly for the smaller diffusion coefficients, and even for $D \geq 10CD^2h^{-1}$, the centre of density for the posteriors still sits substantially above the true value (around 0.26). This is one instance of an overall systematic bias in these estimates which tends to predict higher values of $P_{CC}$ than is actually present, especially when diffusion is small. This is because, when extracellular virus diffuses slowly, it is more likely to result in

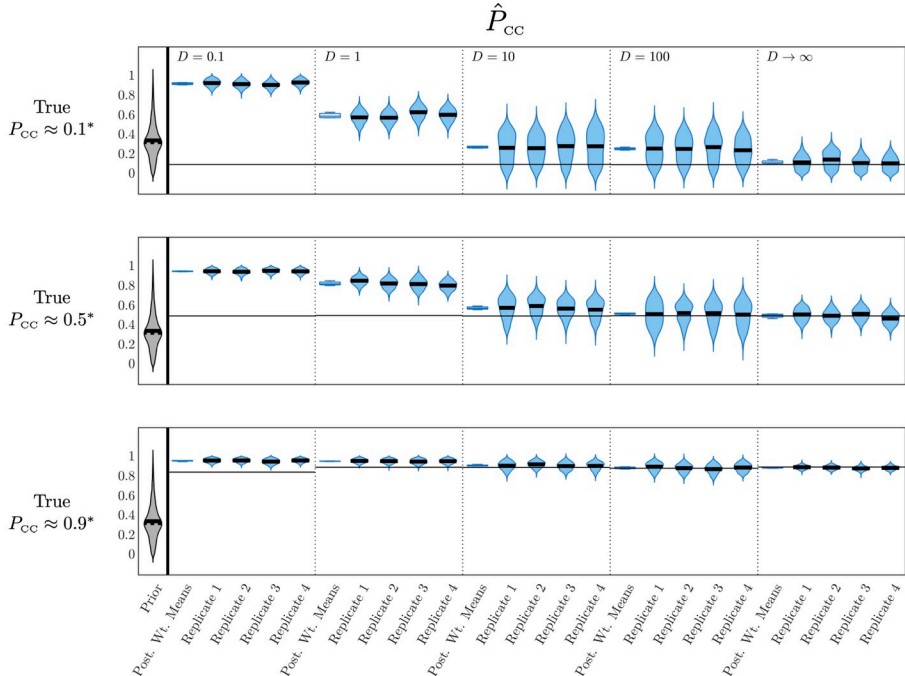

**Fig 6. Fitting data from diffusion–limited viral spread with the spatial model recovers the proportion of cell–to–cell infection for a realistic range of diffusion coefficients.** Prior density and posterior densities from individual replicates for $t_{peak}$ for different values of $D$, the value of the extracellular viral diffusion coefficient used in the extended spatial model to generate observational data. We re–fit using the basic spatial model. For each value of $D$ we also show a boxplot of the distribution of posterior weighted means across all four replicates. We show results for the case where the target values of $\alpha$ and $\beta$ give rise to $P_{CC}$ values of approximately 0.1, 0.5, and 0.9 and $t_{peak}$ of approximately 18h for the specified value of $D$. $\alpha$ and $\beta$ have units of h$^{-1}$ and (TCID$_{50}$/ml)$^{-1}$h$^{-1}$, respectively. *$\alpha$ and $\beta$ values computed from lookup table for relevant diffusion coefficient, ensuring a time of peak infected cell proportion at approximately 18h and the indicated proportion of cell–to–cell infection. $\alpha$ and $\beta$ values for each value of $D$, $t_{peak}$ and $P_{CC}$ used are specified in Table A in S5 Text. $D$ has units of CD$^2$h$^{-1}$, where CD is a cell diameter.

cell–free infections near infection foci which are then mistaken for cell–to–cell infections. When true cell–to–cell infection is rare this effect is exacerbated. While this systemic bias limits the ability of the inference to deduce precise estimates of the actual $P_{CC}$ value in cases where the data is generated using a small diffusion coefficient, it may still be useful in providing an upper bound for this quantity, for instance in the case where $D = 1$CD$^2$h$^{-1}$ and $P_{CC} = 0.1$, where our estimates would at least indicate that cell–to–cell infections are at least not the predominant mode of infection.

The posterior estimates in Fig 6 also have the striking feature that even when accuracy is very low, precision remains very high, and with consistent means across replicates. This property is a consequence of the misspecification of the model used to fit the data, which does *not* include finite diffusion. In S7 Fig we plot the same $\kappa(t)$ trajectories as in Fig 5B–5F, but grouped by $P_{CC}$. S7 Fig demonstrates that, for small and equal cell–to–cell infection proportions ($P_{CC} = 0.1$ or 0.5), the $\kappa(t)$ curve varies substantially for varying values of the diffusion coefficient. Thus, even if, as we predicted in Fig 5G, there is a significant difference between the $\kappa(t)$ curves for different $P_{CC}$ values and a given diffusion coefficient, those curves might be notably different to those for the infinite diffusion case. As such, we might obtain a better fit to the observed $\kappa(t)$ values for an incorrect $P_{CC}$ value. This might explain why the fits in Fig 6 appear to underperform compared to the predicted variation between curves in Fig 5G. Better

estimates could potentially be obtained by refitting the observational data with a model which incorporated finite viral diffusion. However, fitting with such a model would require also fitting the diffusion coefficient $D$, and it is not clear *a priori* the accuracy with which this parameter can be inferred.

## Inference on the prevalence of cell–to–cell infection is robust to smaller samples of the cell sheet

The clustering metric $\kappa(t)$, as we have defined it, relies on sampling every fluorescent cell in the tissue at each observation time and calculating the proportion of its neighbours which are also fluorescent. However, in an experimental setting, it may be impractical if not impossible to observe the fluorescent state of every cell in the target population, especially *in vivo*. We sought to investigate whether approximations of $\kappa(t)$ generated by sampling from subsets of the cell population would be sufficient to allow $\alpha$ and $\beta$—and therefore $P_{CC}$—to be inferred. We did so by carrying out simulation–estimations as in the previous result, but where the clustering metric is now approximated by $\kappa_S(t)$, which is computed by randomly sampling $S$ cells instead of sampling the entire grid. Full details of this adjusted simulation–estimation process are given in Methods "Clustering metric—$\kappa(t)$".

To test the influence of the sample size $S$ on estimation of $P_{CC}$, we performed a series of simulation–estimations on the spatial model using both fluorescence and approximate clustering data for varying sample sizes and target values of $P_{CC}$. These simulation–estimations were conducted using the same methods as in the previous results. We show the results of these simulation–estimations in Fig 7. Here we plot, as in previous figures, weighted posterior

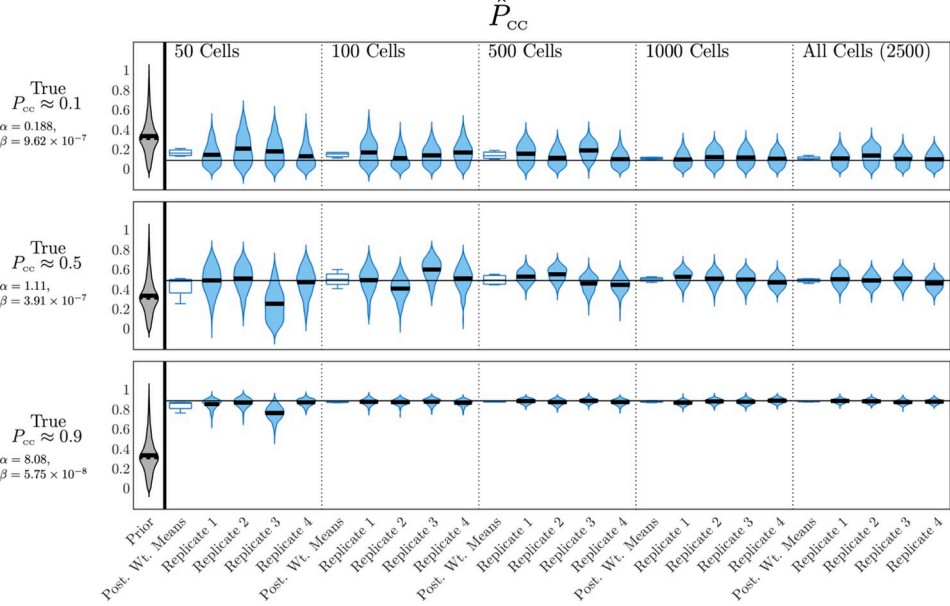

**Fig 7. The prevalence of cell–to–cell infection can be recovered when the fluorescence clustering metric is computed on small samples of the tissue.** Prior density and posterior densities from individual replicates for $P_{CC}$ for different values of $S$, the number of cells sampled to calculate the approximation $\kappa_S(t)$ in fitting. Dashed and solid horizontal lines mark the weighted mean and median values respectively. For each value of $S$ we also show a boxplot of the distribution of posterior weighted means across all four replicates. We show results for the case where the target values of $\alpha$ and $\beta$ give rise to $P_{CC}$ values of approximately 0.1, 0.5, and 0.9 and $t_{peak}$ of approximately 18h. $\alpha$ and $\beta$ have units of $h^{-1}$ and $(TCID_{50}/ml)^{-1}h^{-1}$, respectively.

distributions for $P_{CC}$ for each combination of target parameters and sample size, as well as box plots of the posterior weighted means across replicates in each case. Estimates for $t_{peak}$ are again very precise across all replicates, as is shown in S9 Fig. Fig 7 shows that as the size of the sample becomes smaller and the approximation of $\kappa(t)$ becomes coarser, posterior distributions for $P_{CC}$ become wider and less confident, however, the centre of these distributions is still accurate, as can be seen in the box plots of posterior weighted means, which remain very compact and close to the true value of $P_{CC}$. This is true even for the smallest sample sizes and for any target value of $P_{CC}$. We see that increasing noise due to a reduction in sample size when approximating $\kappa(t)$ does not result in biased estimates of $P_{CC}$, instead, merely a reduction of confidence. By contrast, as we mentioned in the previous result and S1 Text, while an increase in observational noise did lead to an increase in posterior distribution width, it also resulted in individual replicates where $P_{CC}$ estimates were found in reasonably tight, inaccurate distributions. Finally, we also note that, as seen in the previous result, estimation of $P_{CC}$ is far more precise in the case where the target value was higher. Even with the coarsest approximation of the clustering metric, the algorithm correctly identified the $P_{CC}$ in this case with a high degree of precision. This suggests both that high $P_{CC}$ dynamics of the spatial model are particularly distinctive—at least as far as the fluorescence and clustering time series are concerned—but also that only tiny samples of the cell sheet need to be measured in order to precisely infer the value of $P_{CC}$ in this case.

## Discussion

In this work we have conducted a number of simulation experiments to investigate the use of mathematical models in inferring the relative proportions of cell–to–cell and cell–free viral infection, which we summarised via the metric $P_{CC}$: the proportion of infections arising from the cell–to–cell route. We have applied simulation–estimation techniques using Bayesian methods for inference on both an ODE model and a spatially–explicit multicellular model. As much as possible, we aimed to emulate the type and quality of data available experimentally.

In particular, we extracted and attempted to fit time series data on the proportion of fluorescent susceptible cells (that is, initially susceptible cells which have reached, or passed, the productively infected state), following experimental work by Kongsomros and colleagues [6]. We found that this data source was insufficient for inferring $P_{CC}$ from simulation–estimation after observational noise was applied, even when all model parameters aside from those governing the rates of cell–to–cell and cell–free infection were assumed known. This was true for both the ODE and spatial models. By contrast, from the same experiments, *global* metrics of the infection dynamics were very robustly inferred (the exponential growth rate $r$ for the ODE model, and the time of peak infected cell population $t_{peak}$ in the spatial case). This indicates that $P_{CC}$ values can be interchanged while preserving the fluorescent proportion curve—at least as precisely as can be estimated once observational noise is applied—provided $r$ or $t_{peak}$ are held fixed. This suggests that for both the ODE and spatial models, $P_{CC}$ cannot be inferred based on fluorescence data alone. The slight caveat to this claim was our observation that $P_{CC}$ was somewhat well estimated by the spatial model when the true proportion of cell–to–cell infections was high. This was due to the fact that in the spatial model, cell–to–cell infection is forced to spread radially, while cell–free infection is free to spread globally (causing the infected population to grow asymptotically exponentially). Therefore in instances where the global route of infection is almost entirely eliminated, the fluorescent population is forced to grow in a non–exponential manner, which was more easily detected by our inference methods.

We were able to overcome the inability to infer $P_{CC}$ by adding a second set of observational data alongside the fluorescent proportion time series. We did so by introducing a clustering metric $\kappa(t)$, which, given the state of the cell grid in a simulation of the spatial model, measures the mean fraction of fluorescent cells neighbouring each fluorescent cell. Note that since $\kappa(t)$ relies on knowledge of the actual spatial configuration of infection, it is only possible to construct such a metric for a spatially–structured model. We re–ran simulation–estimations on the spatial model, using time series for both the fluorescent cell proportion and $\kappa(t)$ as the observational data, and found that $P_{CC}$ was very well estimated in this case regardless of the target value of $P_{CC}$, however estimates were especially precise when $P_{CC}$ was high.

Since our spatial model assumed uniform extracellular virus (corresponding to infinitely fast viral diffusion) we tested whether the approach outlined here would apply when observational data was obtained with diffusion–limited viral spread. We generated synthetic fluorescence and clustering observational data using an extended spatial model with finite viral diffusion of varying rates and re–estimated $P_{CC}$ and $t_{peak}$ using the misspecified uniform extracellular virus model. Despite not accounting for finite diffusion, the simulation–estimation provided reasonable estimates of both $P_{CC}$ and $t_{peak}$ provided the diffusion coefficient was at least around $10 \text{CD}^2\text{h}^{-1}$, which is lower than the Stokes–Einstein estimate for diffusion of influenza or SARS–CoV–2 virions in body plasma at body temperature [25, 27]. At lower values of the diffusion coefficient ($D = \mathcal{O}(1)\text{CD}^2\text{h}^{-1}$), estimates lose accuracy and incur a substantial bias, but the simulation–estimation may still offer qualitative upper bounds on the prevalence of cell–to–cell infection. It is therefore of great interest what the true value of the extracellular viral diffusion coefficient is under given conditions. Unfortunately, this quantity is not known. Sego and colleagues, for instance, provided a plausible range for the diffusion coefficient of SARS–CoV–2 virions in lung mucus which spanned six orders of magnitude [24]. Reflecting this uncertainty, in our analysis here we have explored a wide range of biologically reasonable diffusion coefficients and obtained useful inferences for a realistic interval of values.

We also found that $P_{CC}$ could still be reliably inferred using the spatial model when the clustering metric $\kappa(t)$ was only coarsely approximated, using a random subset of the cell population. Even at the coarsest approximation we tested—where $\kappa(t)$ was approximated using a sample of only 50 cells—inference of $P_{CC}$ was still reasonably robust, and dramatically improved compared to the case where $\kappa(t)$ was not used at all. These results suggest that even a very rough measure of the spatial distribution of infection is sufficient to deduce the $P_{CC}$ of the underlying system.

One of the limitations to the analysis which we have presented here is the fact that our simulation–estimations have only attempted to fit the parameters governing the rates of infection (that is, $\alpha$ and $\beta$), and assumed perfect prior knowledge of all other model parameters. This prior knowledge is not available when fitting to actual experimental data. There are additional identifiability concerns attached with estimating the other parameters—the cell-free infection rate $\beta$ and extracellular viral production rate $p$, for instance, are well known to only be determined as a product [27, 28]—and it is possible that estimating these additional parameters may introduce further complications in determining $P_{CC}$. Moreover, our work has presented a *practical* identifiability analysis of our model systems, and not a *structural* identifiability analysis. For the sake of simplicity, as well as constraints on computational complexity, we have not carried out a structural identifiability analysis in this work, however this investigation in future will provide further insights into the use of mathematical models in the inference of the prevalence of cell–to–cell infection.

It is worth also briefly remarking on the computational costs associated with parameter estimation using these models. While the ODE model was very efficient to use, inference on the

spatial model was extremely computationally intensive. The computation behind Fig 7, for instance, which comprises 60 individual simulation–estimations, took approximately 13 weeks to complete, with a single typical replicate taking around 24 hours each (running in parallel across eight CPUs (Intel Xeon CPU E5–2683 v4)), while our 150 ODE fits finished in ten days running on four CPUs (AMD EPYC 7702). This is despite using a small $50 \times 50$ grid of cells for the spatial model and only fitting two parameters. The extremely high computational costs associated with these parameter estimations is largely due to the stochastic nature of the spatial model, meaning that many candidate parameter samples which are very close to the true values are randomly rejected. This effect is exacerbated when the noise associated with the model is increased, specifically, when the approximation of $\kappa(t)$ is especially coarse. While recent works in the literature have demonstrated rapid advancements in the speed of simulations, for example, by running on Graphic Processing Units [20] (our code, by contrast, is written in the comparatively slow MATLAB and run on CPUs), the computational costs associated with computing large–scale parameter estimations using the spatial model are not insignificant.

Another important simplification in our approach was our implementation of a global extracellular virus population in the spatial model, rather than a spatially-explicit, diffusing viral population. It is important to clarify here that our use of a global extracellular viral population is based on an assumption of rapid viral transport. This is an important distinction from the modelling literature on HIV (e.g. [1, 11, 12, 17]), where the system can be characterised by well–mixed dynamics since the target cells are also motile. This fact substantially changes the mode of action of the cell–to–cell mechanism and thus also the spatial structure of the infection. For this reason, the methods we have developed here do not extend to HIV infections. We observed that for values of the viral diffusion coefficient far smaller than the Stokes–Einstein estimate ($\mathcal{O}(0.1)\mathrm{CD^2h^{-1}}$), our inference fails. It is highly likely that the increased viscosity of lung mucus and other obstacles *in vivo* are likely to restrict the spread of free virions within the host compared to the Stokes–Einstein estimate [17], and, although the extent of this is unknown, diffusion coefficients in this range have been used by other authors [24, 25]. As such, based on the best quantitative information available, the inference approach outlined here is likely to provide useful estimates of the proportion of cell–to–cell infection, however, should the actual diffusion coefficient turn out to be significantly smaller, this would substantially increase the difficulty of the inference problem. Fitting with a model which accounts for finite viral diffusion could offer an improved fit to the data. We saw in Fig 5G that there is still substantial variation in clustering metrics for changes in $P_{CC}$, even at very low diffusion coefficients. However, such a model would come at a substantially increased computational cost, and would require the diffusion coefficient $D$ to be estimated along with $\alpha$ and $\beta$ (if not also the other parameters of the model), significantly adding to the number of simulation iterations needed to fit the model. It is moreover not clear *a priori* how well the diffusion coefficient would be estimated or how estimates of $P_{CC}$ would be influenced by inaccurate estimation of the diffusion coefficient.

We opted to use fluorescence data as the main data source used in fitting, instead of extracellular viral titre data, which is more typically reported in the experimental virology literature. This is mainly because our work was guided by the results published by Kongsomros and colleagues [6], which reports fluorescent cell proportions as its main metric, but also since we were interested in analysing infection scenarios ranging from the extremes of purely cell-free to purely cell–to–cell, and cell fluorescence data is more relevant to predominantly cell–to–cell infections where cell–free virus has little influence on the dynamics. Furthermore, viral titre observations, as opposed to cell–based observations, do not easily permit the collection of spatial information.

Our work is not the first in the literature to attempt to quantify the relative roles of cell–free and cell–to–cell infection routes. A number of mathematical modelling publications [7, 11, 12, 14, 15, 17, 18], along with experimental works [6, 10] have applied varying models and methods to determine the prevalence of cell–to–cell infection. A common theme among the majority of these works is the use of data collected from infections where one mode of infection is inhibited: either the cell–to–cell mechanism [11–13], or the cell–free mechanism [6, 10, 15]. This approach has substantial limitations. For one, this inhibition process may either restrict or enhance the efficacy of the other mode of infection, either directly or by interrupting the synergistic relationship between the two mechanisms, as we discussed in the Introduction [8, 11, 12]. This approach is moreover limited to *in vitro* settings.

The alternative approach—collecting data from experiments in which both modes of infection are unimpeded—raises additional challenges, but is more robust and, since it requires less invasive experimental intervention, dramatically widens the scope of experiments able to be used for inference. However, earlier estimates of the proportion of infections from the two modes of spread using this data have been subject to substantial uncertainty [7, 17, 18]. Kumberger and collaborators used a spatial model with two modes of infection to generate synthetic global observational data (similar to the fluorescence data we have used here) and attempted to fit it using ODE models [18]. As we have found here, their work suggested that models which (artificially) account for the spatial structure of infection provided better estimates of the prevalence of cell–to–cell spread $P_{CC}$. However, even then, these estimates were still not especially accurate and were subject to systematic biases, even when fitting multiple observational datasets in a single fit. Another study by Imle and colleagues also calibrated an ODE model with two modes of spread to experimental viral load and infected cell count data from an *in vitro* HIV system, and encountered confidence intervals for the proportion of cell–to–cell infection ranging almost all the way from 0–100% [17]. Our work provides context for these findings, offers novel insight on the practical identifiability of $P_{CC}$, and suggests an improved method for determining this quantity. We showed that ODE systems were unable to identify $P_{CC}$, even when fitting data generated by the system itself, and moreover showed that the collection of spatial information, in the form of the clustering metric $\kappa(t)$, was necessary to learn $P_{CC}$, even with a spatial model.

Our hope is that this work provides the foundations for applying mathematical modelling and inference methods to real experimental data in order to accurately quantify the relative roles of cell–free and cell–to–cell spread in real viral infections. The obvious extension to our work here is to apply our methods to experimental data. The data sources we have assumed here—the fluorescent cell proportion time series and the time series for the clustering metric $\kappa(t)$—are readily obtainable (or at least estimable) from model cellular systems. This could be achieved *in vitro* by following standard laboratory methods, and would only require simple staining and imaging techniques [6, 9, 29]. After harvesting and fixing the cell sheet at one of a specified set of observation times, fluorescent cells are easily identified by staining with fluorescent antibodies and imaging the cell sheet. The resulting image could then be processed to compute the fluorescent proportion of the cell population, and to compute or estimate the clustering metric $\kappa(t)$. We do not conduct such an analysis here, preferring instead to leave this for detailed study in a future work. In their study, Kongsomros and colleagues show images only of very small sections of the cell sheet consisting of approximately 10–15 cells, which is insufficient for inference [6]. By contrast, other available experimental images contain very large populations of cells which require automated image processing [30, 31]. Another potential obstacle to analysis of experimental data is in collecting data at a sufficient number of time points. Since time series data of the type assumed here involves destroying the cell sheet at the point of collection, it is expensive to collect data at fine time resolution [6, 9, 31]. We

moreover explored the possible influence of additional observational noise that may be present in experimental data in S2 Text and found that while additional observational noise reduces certainty in predictions of cell–to–cell infection proportions, it does not create systemic biases. These complications influencing the experimental application of our methods here will be explored in future studies.

In brief, this work has explored the identifiability of the relative proportions of cell–free and cell–to–cell infection (the latter of these we termed $P_{CC}$) in two standard models of dual–spread viral dynamics: one ODE model and one spatially–explicit multicellular model. We showed that $P_{CC}$ could not be determined using either model when only the proportion of fluorescent cells was reported. We found that when an additional data source, describing the clustering structure of the infection, was also used for fitting, $P_{CC}$ could be accurately determined using the spatial model. This was the case even when the clustering metric was only approximated using a small sample of the cell sheet, or when the model was fit to observational data with realistic constraints on the diffusion of free virions. Our results imply that some degree of information about the spatial structure of infection is necessary to infer $P_{CC}$. We have demonstrated practically obtainable data types which, combined with experimental collaboration, could lead to more precise and robust predictions of the role of the two modes of viral spread.

## Methods

### An ODE model for dual–spread dynamics

We employ an ODE model which is adapted from a typical model of viral dynamics with two modes of spread [18], which is in turn based on the standard model of viral dynamics [28]. We make the additional inclusion of a latent phase of infection, based on observations from data published by Kongsomros and colleagues [6]. We noticed a delay in the initial uptick of the fluorescent cell time series curve, indicating that cells only become detectably fluorescent once they are productively infected, that is, following the eclipse phase of infection. We tested having both single and multiple latent stages in the model—or equivalently, exponentially and gamma–distributed durations for the eclipse phase—and obtained dramatically improved agreement with the data when we assumed multiple latent stages before cells become detectably fluorescent. This approach is common in representing the eclipse phase of infection in the literature [19, 20]. We arrived at the following form of the model, in ODE form:

$$\frac{dT}{dt} = -\alpha TI - \beta TV, \tag{1}$$

$$\frac{dE^{(1)}}{dt} = \alpha TI + \beta TV - K\gamma E^{(1)}, \tag{2}$$

$$\frac{dE^{(k)}}{dt} = K\gamma (E^{(k-1)} - E^{(k)}), \qquad \text{for } k = 2, 3, ..., K, \tag{3}$$

$$\frac{dI}{dt} = K\gamma E^{(K)} - \delta I, \tag{4}$$

$$\frac{dV}{dt} = pI - cV, \tag{5}$$

where $T$ is the fraction of cells susceptible to infection, $\sum_{i=1}^{K} E^{(i)}$ is the fraction of cells in the

eclipse phase of infection, $I$ is the fraction of cells in the productively infected state, and $V$ is the quantity of extracellular virus. Since we wish to keep track of whether infections come from the cell–to–cell or cell–free infection routes, we incorporate the following subsystem which keeps track of the cumulative proportion of the target population which has become infected via the cell–to–cell mechanism ($F_{\mathrm{CC}}$) or the cell–free mechanism ($F_{\mathrm{CF}}$). We have

$$\frac{dE_{\mathrm{CC}}^{(1)}}{dt} = \alpha TI - K\gamma E_{\mathrm{CC}}^{(1)}, \tag{6}$$

$$\frac{dE_{\mathrm{CC}}^{(k)}}{dt} = K\gamma(E_{\mathrm{CC}}^{(k-1)} - E_{\mathrm{CC}}^{(k)}), \qquad \text{for } k = 2, 3, ..., K, \tag{7}$$

$$\frac{dE_{\mathrm{CF}}^{(1)}}{dt} = \beta TV - K\gamma E_{\mathrm{CF}}^{(1)}, \tag{8}$$

$$\frac{dE_{\mathrm{CF}}^{(k)}}{dt} = K\gamma(E_{\mathrm{CF}}^{(k-1)} - E_{\mathrm{CF}}^{(k)}), \qquad \text{for } k = 2, 3, ..., K, \tag{9}$$

$$\frac{dF_{\mathrm{CC}}}{dt} = \frac{K\gamma}{T_0} E_{\mathrm{CC}}^{(K)}, \tag{10}$$

$$\frac{dF_{\mathrm{CF}}}{dt} = \frac{K\gamma}{T_0} E_{\mathrm{CF}}^{(K)}, \tag{11}$$

where $T_0 = T(0)$ is the initial target cell proportion. The sum of these two quantities,

$$F(t) = F_{\mathrm{CC}}(t) + F_{\mathrm{CF}}(t), \tag{12}$$

is the cumulative proportion of the cell population which has become infected through either mechanism, which we take to be equivalent to the proportion of fluorescent cells as observed in Kongsomros et al. [6]. The assumption that cells remain fluorescent even after they die (over the time scale of interest) is justified by the observation that in Kongsomros et al. fluorescent proportions were observed to saturate at 100% at later times in their experiments.

Throughout this work, we will assume fixed values of the parameters $K$, $\gamma$, $\delta$, $p$, and $c$, as specified in Table 1. These parameters were obtained by running a Bayesian parameter estimation for the form of the ODE model as defined above against fluorescent cell time series data in Kongsomros et al. [6], and selecting one particular posterior sample at random. We sketch this parameter estimation process in S6 Text. These values were selected simply to be indicative of the realistic range of values for these parameters and are sufficiently realistic for the purposes of this work. In each case we initiate the infection by setting $T(0) = 0.99$, $I(0) = 0.01$ and the remaining compartments to zero.

**Table 1. Fixed parameters used in our simulations.**

| Description | Symbol | Value and Units |
|---|---|---|
| Number of delay compartments | K | 3 |
| Eclipse cell activation rate | $\gamma$ | $3.366934 \times 10^{-1} \mathrm{h}^{-1}$ |
| Death rate of infected cells | $\delta$ | $8.256588 \times 10^{-2} \mathrm{h}^{-1}$ |
| Extracellular virion production rate | $p$ | $1.321886 \times 10^{6} (\mathrm{TCID}_{50}/\mathrm{ml}) \mathrm{h}^{-1}$ |
| Extracellular virion clearance rate | $c$ | $4.313531 \times 10^{-1} \mathrm{h}^{-1}$ |

## A multicellular spatial model for dual–spread dynamics

It is straightforward to adapt this system of ODEs into a spatially–structured multicellular model, that is, a model which tracks the dynamics of a finite number of discrete cells which each occupy some specified region of space and at any given point in time, may be in one of a set of cell states [32, 33]. Suppose we model the dynamics of a population of $N$ cells. We associate with each of these cells an index $i \in \{1, 2, \ldots, N\}$, and a cell state at time $t$ given by $\sigma_i(t)$, where the possible cell states correspond to the compartments of the ODE system, including the implicit dead cell compartment. That is, for any cell $i$, $\sigma_i(t) \in \{T, E, I, I^\dagger\}$, representing the target, eclipse, infected and dead state respectively.

We consider a two–dimensional sheet of cells with hexagonal packing of cells and periodic boundary conditions in both the $x$ and $y$ directions, such that each cell has precisely six neighbours. This packing reflects the arrangement of cells in real epithelial monolayers and has the practical benefit that all adjacent cells are joined via a shared edge, avoiding any complications associated with corner–neighbours. Throughout this work we use a $50 \times 50$ grid of cells.

Below, we define both a basic and an extended spatial model. Throughout this work, we use the basic model for inference. The extended model is used only in specified instances for the generation of observational data. For the *basic* spatial model, following other authors [15, 17, 18, 34], we make the simplifying assumption that the dispersal of free virions over the computational domain is fast, and that the extracellular viral distribution can therefore be considered approximately uniform. As such, the equation for $V$ in our spatial model changes only in notation from Eq (5):

$$\frac{dV}{dt} = p \sum_{i=1}^{N} \frac{\mathbb{1}_{\{\sigma_i(t)=I\}}}{N} - cV. \tag{13}$$

As such, cell–free infection is considered a spatially *global* mode of spread in our spatial model. By contrast, following results from the biological literature, we assume that cell–to–cell spread is a spatially *local* mechanism [3, 6]. As such, we assume that the probability of cell–to–cell infection in the spatial model depends not on the global proportion of infected cells as in Eq (1), but rather the proportion of a cell's neighbours which are infected. Specifically, if we denote by $v(i)$ the set of indices of the cells neighbouring cell $i$, and by $n_{\text{neighbours}} = 6$ the fixed number of neighbours a cell can have, the probability of cell $i$ becoming infected by cell–to–cell infection over a given time period depends on the term $\sum_{j \in v(i)} (\mathbb{1}_{\{\sigma(j)=I\}})/n_{\text{neighbours}}$. Combining these two mechanisms, we obtain the following transition probability for target cell $i$ to become (latently) infected over some time interval $\Delta t$:

$$P(\sigma_i(t + \Delta t) = E | \sigma_i(t) = T) = 1 - \exp\left(-\left(\alpha \sum_{j \in v(i)} \frac{\mathbb{1}_{\{\sigma_j(t)=I\}}}{n_{\text{neighbours}}} + \beta V\right)\Delta t\right). \tag{14}$$

We also define an *extended* spatial model which relaxes the assumption that extracellular viral transport is approximately instantaneous. To do so, we assume that extracellular viral density obeys linear diffusion in the environment with diffusion coefficient $D$ $\text{CD}^2\text{h}^{-1}$ (where CD is a cell diameter, defined as the constant distance between cell centres). If we denote by $S_i$ the region of space (in $\mathbb{R}^2$) occupied by cell $i$, we assume that extracellular virus is secreted by each productively infectious cell $j$ uniformly over $S_j$, and that any susceptible cell $k$ can become

infected by the extracellular viral density in $S_k$. Specifically, we have for the virus equation

$$\frac{\partial V}{\partial t} = p \sum_{i \in \mathcal{I}(t)} \frac{\mathbb{1}_{\{\mathbf{x} \in S_i\}}}{|S_i|} - cV + D\nabla^2 V, \tag{15}$$

and, correspondingly, the transition probability for infection becomes

$$P(\sigma_i(t + \Delta t) = E | \sigma_i(t) = T) = 1 - \exp\left( -\left( \alpha \sum_{j \in v(i)} \frac{\mathbb{1}_{\{\sigma_j(t) = I\}}}{n_{\text{neighbours}}} + \beta N \int_{S_i} V d\mathbf{x} \right) \Delta t \right). \tag{16}$$

We numerical solve the virus PDE using a implicit–explicit Finite Difference Method using nodes at each of the cell centres. For further details, refer to S5 Text.

For the eclipse phase, instead of implementing transition probabilities for each $E^{(k)}$, for computational simplicity we instead sample a latent phase duration from its probability distribution at the time a cell first enters the eclipse state. That is, if we write $t_i^E = \min\{t : \sigma_i(t) = E\}$ for the time at which cell $i$ enters the eclipse state, and $t_i^I = \min\{t : \sigma_i(t) = I\}$ for the time at which cell $i$ enters the productively infected state, we have

$$t_i^I = t_i^E + \tau_i, \tag{17}$$

where

$$\tau_i \sim Gamma\left( K, \frac{1}{K\gamma} \right). \tag{18}$$

The remaining compartments are easily described by simple transition probabilities.

$$P(\sigma_i(t + \Delta t) = I | \sigma_i(t) = E) = 1 - \exp(-K\gamma\Delta t), \tag{19}$$

$$P(\sigma_i(t + \Delta t) = I^\dagger | \sigma_i(t) = I) = 1 - \exp(-\delta\Delta t). \tag{20}$$

Together with appropriate initial and boundary conditions, Eqs (13), (14) and (17)–(20) define the basic spatial model, and Eqs (15)–(20) define the extended spatial model. Following equivalent initial conditions as for the ODE model, in both the basic and extended case we initiate infection by randomly selecting 1% of the cell sheet to be initially infected, and the remainder of the sheet to be susceptible to infection. We use periodic boundary conditions in $x$ and $y$. In Fig 3A we show a schematic of the model as well as the layout of the cell grid. This is not a novel model: this model structure, or slight variations thereof, has been used in a number of recent publications describing infection dynamics with two modes of viral spread and has become somewhat of a standard approach in the field in recent years [7, 14, 15, 20].

As with the ODE model, we can additionally keep track of the cumulative proportion of infections arising from each mode of infection individually in the spatial model. In addition to the overall probability of infection in Eq (14), we can compute a probability of infection by each mode of spread individually as follows. Using the same Poisson process argument as above, the probability of cell–to–cell infection of cell $i$ *not* taking place over the time interval $[t, t + \Delta t)$ is given by

$$P(\mathbf{E}_i^{\text{CC}} \notin [t, t + \Delta t)) = \exp\left( -\alpha \sum_{j \in v(i)} \frac{\mathbb{1}_{\{\sigma_j(t) = I\}}}{n_{\text{neighbours}}} \Delta t \right), \tag{21}$$

and the probability of cell–free infection of cell $i$ not occurring over the same time interval is

given by

$$P(\mathbf{E}_i^{\mathrm{CF}} \notin [t, t + \Delta t)) = \exp(-\beta V \Delta t), \tag{22}$$

for the basic model, and

$$P(\mathbf{E}_i^{\mathrm{CF}} \notin [t, t + \Delta t]) = \exp\left(-\beta \int_{S_i} V d\mathbf{x} \Delta t\right), \tag{23}$$

for the extended model, where $\mathbf{E}_i^{CC}$ and $\mathbf{E}_i^{CF}$ are the events of a cell–to–cell infection and a cell–free infection occurring at cell $i$ respectively. Note that we have to account for the fact that while, mathematically, both events may occur in the time interval $[t, t + \Delta t)$, we need to assign a unique mode of transmission to each infection. We do so as follows. The following calculation is also derived in work by Blahut and colleagues [15]. If we write $m(i) \in \{CC, CF\}$ for the mode of infection of cell $i$, then at the time of infection of cell $i$—that is, when $t = t_i^E$—we compute the probability of each individual mode of transmission as follows:

$$P(m(i) = CC) = \frac{1 - P(\mathbf{E}_i^{\mathrm{CC}} \notin [t, t + \Delta t))}{2 - P(\mathbf{E}_i^{\mathrm{CC}} \notin [t, t + \Delta t)) - P(\mathbf{E}_i^{\mathrm{CF}} \notin [t, t + \Delta t))}, \tag{24}$$

and

$$P(m(i) = CF) = \frac{1 - P(\mathbf{E}_i^{\mathrm{CF}} \notin [t, t + \Delta t))}{2 - P(\mathbf{E}_i^{\mathrm{CC}} \notin [t, t + \Delta t)) - P(\mathbf{E}_i^{\mathrm{CF}} \notin [t, t + \Delta t))}. \tag{25}$$

In our model, therefore, when an infection is detected, we draw a random number $p \sim Uniform(0, 1)$, and if $p < P(m(i) = CC)$, we designate the infection a cell–to–cell infection, otherwise, it is considered a cell–free infection. We use a similar calculation to assign the viral lineage associated with an infection, which we used to construct the colouring of cells in Fig 3D, which we stipulate in full in S3 Text. The quantities $F_{CC}$ and $F_{CF}$ can easily be calculated for the spatial model as

$$F_{\mathrm{CC}}(t) = \frac{\sum_{i=1}^{N} \mathbb{1}_{\{m(i)=CC\}} \mathbb{1}_{\{t_i^I \in [0,t]\}} \mathbb{1}_{\{\sigma_i(0)=T\}}}{\sum_{i=1}^{N} \mathbb{1}_{\{\sigma_i(0)=T\}}}, \tag{26}$$

$$F_{\mathrm{CF}}(t) = \frac{\sum_{i=1}^{N} \mathbb{1}_{\{m(i)=CF\}} \mathbb{1}_{\{t_i^I \in [0,t]\}} \mathbb{1}_{\{\sigma_i(0)=T\}}}{\sum_{i=1}^{N} \mathbb{1}_{\{\sigma_i(0)=T\}}}, \tag{27}$$

which allow us to keep track of the count of each type of infection event throughout simulations of the spatial model. As before, we define

$$F(t) = F_{\mathrm{CC}}(t) + F_{\mathrm{CF}}(t). \tag{28}$$

## Metrics

**Proportion of infections from the cell–to–cell route—$P_{\mathrm{CC}}$.** We introduce the quantity $P_{\mathrm{CC}}$ to denote the proportion of infections arising from the cell–to–cell route. This is calculated by keeping track of the cumulative proportion of the target cell population which becomes infected by either infection mechanism over time. At long time—once the infection has essentially run its course—we compute $P_{\mathrm{CC}}$ as the fraction of the total infections which

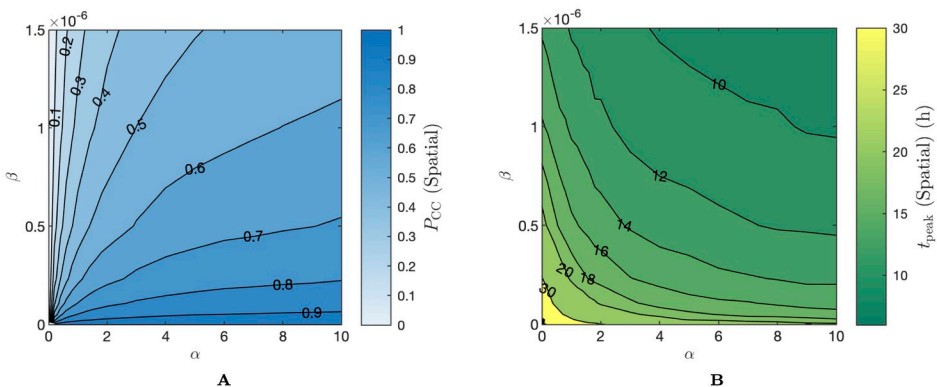

**Fig 8. $P_{CC}$ and $t_{peak}$ contour maps on $\alpha$–$\beta$ space for the spatial model.** Contour maps for (A) $P_{CC}$, and (B) $t_{peak}$. $\alpha$ and $\beta$ have units of $h^{-1}$ and $(TCID_{50}/ml)^{-1}h^{-1}$, respectively.

occurred via cell–to–cell infection. Using $F_{CC}$ and $F_{CF}$ as we have defined them, we have

$$P_{CC} = \lim_{t \to \infty} \frac{F_{CC}(t)}{F_{CC}(t) + F_{CF}(t)}. \tag{29}$$

In Fig 1C we show an illustration of this calculation more generally.

$P_{CC}$ quantifies the relative weight of the cell–to–cell route of infection and is therefore our target for estimation in this work. Its definition is general and is not specific to any particular model structure. $P_{CC}$ cannot be directly calculated in closed form directly from the model parameters. Instead, we repeatedly simulate our model using parameters sampled from $\alpha$–$\beta$ space and compute $P_{CC}$ in order to construct lookup tables. In Fig 1D, we plot a contour map of $P_{CC}$ values for the ODE model in $\alpha$–$\beta$ space. In the case of the spatial model, we accounted for the inherent stochasticity of the model by running 20 simulations of the model at each $(\alpha, \beta)$ pair in the lookup table and kept track of mean $P_{CC}$ values. The associated contour map for the spatial model is shown in Fig 8A. Contour plots were generated by computing contours over the lookup table using MATLAB's `contourf` function. We interpolate between values on the lookup table by constructing spline fits along $\alpha$ and $\beta$ contours.

**Exponential growth rate—$r$.**   A second quantity, which describes the *overall* rate of infection spread is the exponential growth rate $r$. This quantity, related to the basic reproduction number $\mathcal{R}_0$, is well–established in the theory of epidemiological and virus dynamical models and has the property that, for small $t$, we have $I(t) \approx I_0 e^{rt}$ [21, 22, 35, 36]. The exponential growth rate for the ODE model can be readily computed by linearising the ODE system about the infection free steady state and finding the dominant eigenvalue of the resulting system [22, 36]. For our model, we obtain the following explicit definition:

$$r = \max\{\lambda \ : \ g(\lambda) = 0\}, \tag{30}$$

where

$$g(\lambda) := \left(1 + \frac{\lambda}{K\gamma}\right)^K (c + \lambda)(\delta + \lambda) - (\alpha(c + \lambda) + \beta p).$$

**Time to peak infected cell population—$t_{peak}$.**   The exponential growth rate $r$ relies on asymptotically exponential behaviour of the infected proportion curve. However, for the

spatial model, especially in instances where infections spread mainly locally—that is, through the cell–to–cell route—the infected proportion curve does not grow exponentially. For the spatial model, therefore, $r$ is not well-defined. We instead use the time of the peak infected cell proportion, which we label as $t_\text{peak}$, as an alternative measure of the overall growth behaviour of the infected population. As with $P_\text{CC}$, this quantity is not easily approximated *a priori*, therefore we also compute lookup tables in $\alpha$–$\beta$ space for this quantity. We show the contour map of $t_\text{peak}$ on $\alpha$–$\beta$ space in Fig 8B. Contour plots were generated by computing contours over the lookup table using MATLAB's `contourf` function.

The time of the peak infected cell population is a quantity that is not typically experimentally observable, whereas a quantity like the time of peak viral load is comparatively much easier to measure in an experimental context. However, we opt to use the latter metric, since this is a meaningful metric of the model regardless of the mechanism of infection spread. Even in a scenario where all infections in the model arise from the cell–to–cell route (i.e. $P_{cc} = 1$) the time of peak infected population remains a relevant as a measure of the overall rate of infection progression, where the time of the peak extracellular viral load is far less meaningful here. In any event, for our purposes in this work, $t_\text{peak}$ is used simply to illustrate a quantity which represents the overall rate of infection spread in a model simulation, and a quantity which we observe to be preserved between accepted samples of our simulation–estimation (at least when clustering data are not used). This choice of metric does not diminish the relevance of our analysis to experimental application.

**Clustering metric—$\kappa(t)$ (and approximation—$\kappa_S(t)$).**   Given a cell grid where we denote by $\mathcal{F}(t)$ the set of cells which are fluorescent, we compute for each fluorescent cell $i \in \mathcal{F}(t)$ the quantity $k_i(t)$, which is the proportion of the neighbours of cell $i$ which are also fluorescent. We then define $\kappa(t)$ as the mean of the $k_i(t)$s. We have:

$$k_i(t) = \sum_{j \in v(i)} \frac{\mathbb{1}_{\{j \in \mathcal{F}(t)\}}}{|v(i)|}, \tag{31}$$

and

$$\kappa(t) = \frac{1}{N}\sum_{i=1}^{N} k_i(t). \tag{32}$$

We compute $\kappa(t)$ over time $t$ to form a time series. In Fig 3C and 3G, we show an example of computing fluorescent neighbour proportions, and plot example $\kappa(t)$ time series for three parameter pairs, corresponding to $P_\text{CC}$ values of 0.1, 0.5, and 0.9. Fig 3G, shows that, unlike with the fluorescent cell time series, there is substantial variation in the $\kappa(t)$ curves with changing $P_\text{CC}$.

$\kappa(t)$ has the property that when it is near zero, fluorescent cells are mostly isolated and the infection is very diffuse, and when it is near one, fluorescent cells are generally found in clusters, indicating that the infection is very compact. In principle, $\kappa(t)$ could be computed or estimated in experimental settings with the use of fluorescence imaging of the cell sheet, samples of which can be found in works by Kongsomros *et al.* and Fukuyama *et al.* [6, 31].

We modify the definition of $\kappa(t)$ to define the approximation $\kappa_S(t)$ as follows. Given a grid of $N$ cells at time $t$, of which the fluorescent population is given by $\mathcal{F}(t)$ as before, we draw $S \leq N$ cells without replacement and call the set of sampled cells $\mathcal{S}(t)$. For each sampled cell $i$, if $i \in \mathcal{F}(t)$, we compute $k_i(t)$, and then compute the approximate clustering metric $\kappa_S(t)$ as the

mean of the computed $k_i(t)$s, that is, if $|\mathcal{S}(t) \cap \mathcal{F}(t)| \neq 0$, then

$$\kappa_S(t) = \sum_{i \in \mathcal{S}(t) \cap \mathcal{F}(t)} \frac{k_i(t)}{|\mathcal{S}(t) \cap \mathcal{F}(t)|}.$$

Note that $k_i(t)$ is defined as above. That is, for each sampled cell $i$, we still compute $k_i(t)$ from that cell's neighbours, which may not be in the sampled set $\mathcal{S}(t)$. In the event that no fluorescent cells are sampled (that is, $|\mathcal{S}(t) \cap \mathcal{F}(t)| = 0$), we define $\kappa_S(t) = 0$.

## Simulation–estimation

Throughout this work we conduct a series of simulation–estimation experiments to explore what can be learned about the roles of the two modes of viral spread based on observed model outputs. We outline here the general framework of this process.

For both the ODE and the spatial model, we begin by drawing a set of target values for the infection parameters $\alpha$ and $\beta$. As mentioned above, the values of the other model parameters are considered fixed and known. We then simulate the chosen model using these parameter values, and apply an observational model $f(\cdot)$ to its output to generate a set of observed data $\mathcal{D}$. The observational model $f$ is designed to simulate the noise incurred in actual experiments. Throughout this work, we focus especially on the observed *fluorescent cell proportion* over time, since this is the main source of data reported by Kongsomros and colleagues [6].

For the ODE model, we obtain the observed fluorescent cell proportion $\mathcal{D}^{\text{ODE}}$ by computing the true fluorescent cell time series $F(t)$ (defined in Eq (12)) at each of a series of observation times, converting this proportion to a count of fluorescent cells and applying negative binomial noise. The negative binomial distribution reflects the observed error structure in [6], which is constructed from overdispersed count data. We assume some vector of observation times $\mathbf{t} = \{t_1, t_2, \ldots t_m\}$ and define

$$\mathcal{D}^{\text{ODE}} = f^{\text{ODE}}(F(t); \mathbf{t}, \phi, N_{\text{sample}}) = \left( \frac{1}{N_{\text{sample}}} \right) \cdot \{\mathcal{D}^{\text{ODE}}_1, \mathcal{D}^{\text{ODE}}_2, ..., \mathcal{D}^{\text{ODE}}_m\}, \tag{33}$$

where

$$\mathcal{D}^{\text{ODE}}_i \sim \textit{Negative Binomial}(N_{\text{sample}}F(t_i), \phi),$$

for $i = 1, 2, \ldots, m$, where $N_{\text{sample}}F(t_i)$ and $\phi$ are the mean and dispersion parameter respectively of $\mathcal{D}_i$. $N_{\text{sample}}$ is the number of cells measured for fluorescence, in a sense the size of the cell population. $\text{Var}[\mathcal{D}_i] = N_{\text{sample}}F(t_i) + [N_{\text{sample}}F(t_i)]^2/\phi$ for all $i = 1, 2, \ldots m$. In Fig 9B, we show an illustration of this observation process. The curve shown in blue is the true fluorescent proportion curve $F(t)$. At each of the observation times, indicated with dots, we apply noise about the true value.

After obtaining observed data $\mathcal{D}^{\text{ODE}}$, we then re–estimate $\alpha$ and $\beta$ using Bayesian methods. We assume uniform prior distributions

$$\pi_\alpha(\alpha) = \begin{cases} 1/\alpha_{\text{max}}, & \alpha \in [0, \alpha_{\text{max}}], \\ 0, & \text{otherwise}, \end{cases} \tag{34}$$

$$\pi_\beta(\beta) = \begin{cases} 1/\beta_{\text{max}}, & \beta \in [0, \beta_{\text{max}}], \\ 0, & \text{otherwise}, \end{cases} \tag{35}$$

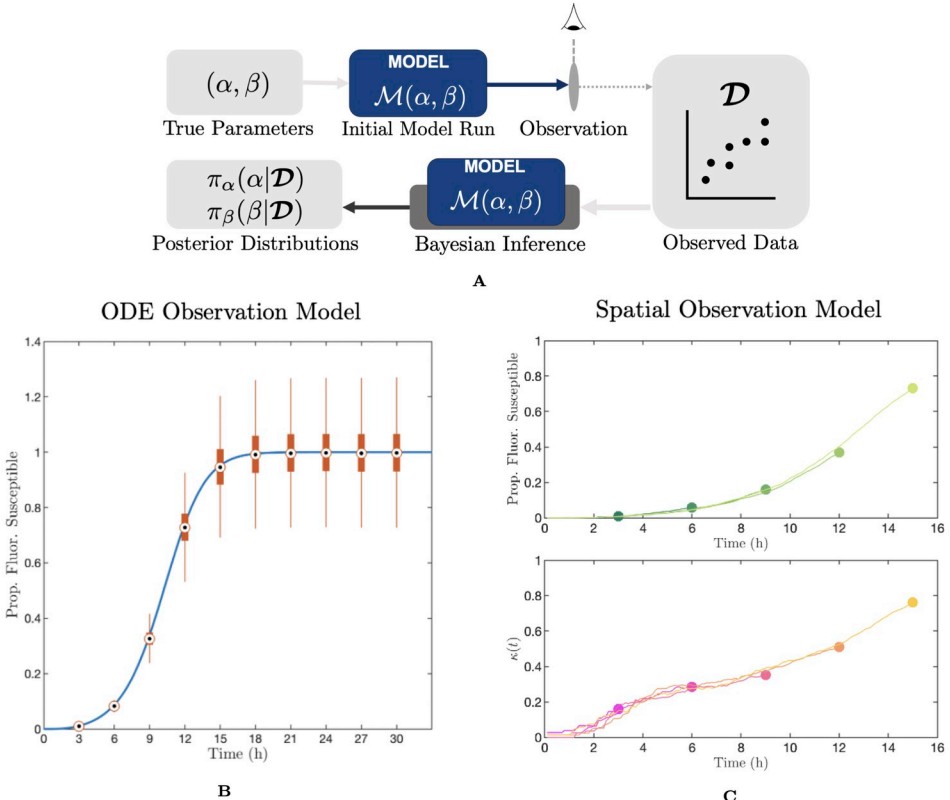

**Fig 9. Schematics of the fitting process and observational models.** (A) Schematic of the simulation–estimation process. (B)–(C) Observation model for fluorescent proportion of susceptible cells for (B) the ODE model and (C) the spatial model (first five points shown). In the spatial case we also show the observation model for the clustering metric $\kappa(t)$. For the ODE model, we sample the true fluorescent proportion curve at a series of time points (shown in blue), then observe a value based on a negative binomial distribution centred on the true value (box plot of the distribution shown in orange). Here, $\phi = 10^2$. For the spatial model, we run independent iterations of the stochastic model and observe one point from each. Note that additional observational noise can be applied to these data points, as we explore in S2 Text.

with $\alpha_{\max} = 2.5\mathrm{h}^{-1}$, $\beta_{\max} = 2 \times 10^{-6}(TCID_{50}/ml)^{-1}h^{-1}$. We re–estimate $\alpha$ and $\beta$ using No U–Turn Sampling (NUTS) Markov Chain Monte Carlo (MCMC) methods with a negative binomial likelihood

$$\mathcal{D}_i^{\mathrm{ODE}} \sim Negative\ Binomial(N_{\mathrm{sample}}\hat{F}(t_i), \phi), \tag{36}$$

for $i = 1, 2, \ldots, m$, where $\hat{F}(t)$ is the fluorescent proportion time series estimated by simulating the ODE model using samples $\hat{\alpha}$ and $\hat{\beta}$. For each estimation we use four chains seeded with random initial values and draw 2000 samples for each, including 200 burn–in samples. We assume $N_{\mathrm{sample}} = 2 \times 10^5$, which was the number of cells used in the experiments in [6].

For the spatial model, we have two sources of observational data: both the fluorescent proportion time series and the clustering metric $\kappa(t)$. Since the system is inherently stochastic, we do not add additional external noise, instead, we aim to emulate the experimental process whereby the fluorescent proportion of a cell population (and consequently the clustering metric) cannot be observed without destroying, or at least disrupting, the cell sheet. We implement this by sampling our observations from $m$ independent simulations of the model.

That is, if we have observation times $\mathbf{t} = \{t_1, t_2, \ldots, t_m\}$ and $m$ true fluorescence and clustering time series from independent simulations of the spatial model, $\mathbf{F}(t) = \{F_1(t), F_2(t), \ldots, F_m(t)\}$ and $\mathbf{K}(t) = \{\kappa_1(t), \kappa_2(t), \ldots, \kappa_m t\}$ respectively, we generate the two sets of observational data,

$$\mathcal{D}_{\text{fluoro}}^{\text{spatial}} = f^{\text{spatial}}(\mathbf{F}(t); \mathbf{t}) = \{F_1(t_1), F_2(t_2), \ldots F_m(t_m)\}, \tag{37}$$

$$\mathcal{D}_{\text{cluster}}^{\text{spatial}} = f^{\text{spatial}}(\mathbf{K}(t); \mathbf{t}) = \{\kappa_1(t_1), \kappa_2(t_2), \ldots \kappa_m(t_m)\}. \tag{38}$$

We show a demonstration of this observation process in Fig 9C.

Due to the stochasticity of the system, we use Approximate Bayesian Computation (ABC) to re–estimate $\alpha$ and $\beta$ for the spatial model. In particular, we adapt the Population Monte Carlo (PMC) method introduced by Toni and collaborators [37] and revised by others [38, 39]. We sketch this method in pseudocode in Algorithm 1.

**Algorithm 1** PMC algorithm for parameter estimation using the spatial model—fluorescence and clustering data

```
Input: Model M(α,β), prior distributions for target parameters π_α(α)
and π_β(β), target number of particles N_P, number of generations G, ref-
erence data D_fluoro^spatial and D_cluster^spatial, distance metrics d_fluoro(·,·) and d_cluster(·,·),
perturbation kernel K(·|·), initial acceptance proportion p_0,accept,
threshold tightening parameter q.
Output: Weighted samples from the posterior distributions
π̂_α(α|D_fluoro^spatial, D_cluster^spatial), π̂_β(β|D_fluoro^spatial, D_cluster^spatial).

Rejection sampling
for i = 1,2,...,⌈N_P/p_0,accept⌉ do
  Randomly draw α̂_i and β̂_i from π_α(α) and π_β(β), respectively.
  Obtain the model output using these parameters,
    {D̂_fluoro^spatial,(i), D̂_cluster^spatial,(i)} = M(α̂_i, β̂_i).
  Compute the distance between model output and reference data
    ϵ_fluoro^i = d_fluoro(D̂_fluoro^spatial,(i), D_fluoro^spatial) and ϵ_cluster^i = d_cluster(D̂_cluster^spatial,(i), D_cluster^spatial).
end for
n_opt found ← 0, T ← 0
while n_opt found < N_P do
  T ← T + 1, define I_1, I_2, ..., I_n_opt found, as the indices i in the smallest T
      values of the ϵ_fluoro^i s and the smallest T values of the ϵ_cluster^i s.
end while
for j = 1, 2, ..., N_P do
  Set P_j = (α̂_I_j, β̂_I_j).
  Set w_j = 1/N_P.
end for
P = {P_1, P_2, ..., P_N_P} is the initial particle population. w = {w_1, w_2, ..., w_N_P} is
the initial weight vector. Set the distance thresholds ϵ_fluoro^D and ϵ_cluster^D
as the q^th quantile of the ϵ_fluoro^i s and ϵ_cluster^i s respectively.

Importance sampling
for g = 1, 2, ..., G do
  Set number of accepted particles N_accepted ← 0
  While N_accepted < N_P do
    Randomly draw a particle P_j with probability w_j.
    Perturb particle by the kernel K(·|P_j) to obtain a new sample (α̂, β̂).
    Obtain the model output using these parameters,
      {D̂_fluoro^spatial,(i), D̂_cluster^spatial,(i)} = M(α̂_i, β̂_i).
```

```
      Compute the distance between model output and reference data
          ε_fluoro^i = d_fluoro(𝒟̂_fluoro^{spatial,(i)}, 𝒟_fluoro^spatial) and ε_i^cluster = d_cluster(𝒟̂_cluster^{spatial,(i)}, 𝒟_cluster^spatial).
      if ε_fluoro^i < ε_fluoro^D and ε_cluster^i < ε_cluster^D then
          Set N_accepted ← N_accepted + 1 and 𝒫_{N_accepted}^next = (α̂_i, β̂_i).
      else
          Return to start of while.
      end if
  end while
  for i = 1, 2, ..., N_P do
      Set w_i^{*,next} = w_i / ∑_{j=1}^{N_P} K(𝒫_i^next|𝒫_j)w_j
  end for
  Set 𝒫 ← {𝒫_1^next, 𝒫_2^next, ..., 𝒫_{N_P}^next},  w ← (1/∑_{i=1}^{N_P} w_i^{*,next}) · {w_1^{*,next}, w_2^{*,next}, ..., w_{N_P}^{*,next}}
  Set the distance thresholds ε_fluoro^D and ε_cluster^D as the q^{th} quantile of the
      ε_fluoro^i s and ε_cluster^i s respectively.
 end for
```

In our case, the model $\mathcal{M}(\hat{\alpha}, \hat{\beta})$ is simply the time series $F(t)$ obtained by a single simulation of the spatial model with parameters $\alpha = \hat{\alpha}$ and $\beta = \hat{\beta}$, and evaluated at time points $\mathbf{t}$, the vector of time points at which the reference data $\mathcal{D}$ is obtained. We again use the uniform prior distributions in Eqs (34) and (35), although now with $\alpha_{max} = 10\text{h}^{-1}$, $\beta_{max} = 1.5 \times 10^{-6}(\text{TCID}_{50}/\text{ml})^{-1}\text{h}^{-1}$. For the perturbation kernel, we use the following definition proposed by Beaumont and colleagues: [38]

$$K(\mathcal{P}_k^*|\mathcal{P}_i) = \Phi(\mathcal{P}_k^*; \mathcal{P}_i, 2\Sigma), \tag{39}$$

where $\Phi(\mathbf{x}; \mu, \sigma^2)$ is a multivariate normal and $\Sigma$ is the empirical covariance matrix of the particle population $\{\mathcal{P}_1, \mathcal{P}_2, ..., \mathcal{P}_{N_p}\}$, using their weights $\{w_1, w_2, ..., w_{N_p}\}$. For the other parameters of the algorithm, we set $N_P = 500$, $G = 5$, $p_{0,\text{accept}} = 0.3$, and $q = 0.5$. We use euclidean distance for the distance metric $d$. For the case where we attempt only to estimate $\alpha$ and $\beta$ using the spatial model and fluorescence data only, we slightly simplify the fitting process. We apply the same observational model, outlined in Eq (37), to the fluorescence data, and use a slightly simplified version of the PMC method to refit $\alpha$ and $\beta$. We provide full details in S7 Text.

## Supporting information

**S1 Fig. ODE model under varying observational noise.** (A) Prior density and posterior densities from individual replicates for $P_{\text{CC}}$ at different levels of observational noise. At each level of noise we also show a box plot of the distribution of posterior medians across all replicates. There are ten replicates in total at each level of noise, of which we display four. The highlighted segment is the level of noise used in the main text. (B) Same as (A), but showing estimates for $r$. (C)–(F) Indicative observed data compared to true fluorescence time series for each value of the dispersion parameter $\phi$ used in (A) and (B). Here $\alpha = 1.09\text{h}^{-1}$, $\beta = 7.20 \times 10^{-7}(\text{TCID}_{50}/\text{ml})^{-1}\text{h}^{-1}$, with $P_{\text{CC}} \approx 0.5$.
(TIFF)

**S2 Fig. Spatial model under varying (artificial) observational noise.** (A) Prior density and posterior densities from individual replicates for $P_{\text{CC}}$ at different levels of observational noise. At each level of noise we also show a box plot of the distribution of posterior medians across all replicates. There are four replicates at each level of noise. The highlighted segment is the level of noise used in the main text (which in this case has no artificial observational noise beyond the inherent stochasticity of the model, as explained in the main text). (B) Same as (A), but showing

estimates for $t_{\text{peak}}$. Here $\alpha = 1.11\text{h}^{-1}$, $\beta = 3.91 \times 10^{-7}(\text{TCID}_{50}/\text{ml})^{-1}\text{h}^{-1}$, with $P_{\text{CC}} \approx 0.5$.
(TIFF)

**S3 Fig. Scatter plots for accepted posterior samples for the ODE model.** Scatter plot of accepted posterior samples in $\alpha$–$\beta$ space for a fit to fluorescence data where the true $P_{\text{CC}} \approx 0.1$, 0.5, 0.9 and fixed $r$ using the ODE model, as presented in Fig 2 of the main article.
(TIFF)

**S4 Fig. $\alpha$ and $\beta$ marginal posterior distributions—ODE model.** Posterior and prior distributions for $\alpha$ and $\beta$ for simulation–estimations with the ODE model presented in Fig 2 of the main article.
(TIFF)

**S5 Fig. $\alpha$ and $\beta$ marginal posterior distributions—spatial model with clustering data.** Posterior and prior distributions for $\alpha$ and $\beta$ for simulation–estimations with the spatial model (with the clustering metric) presented in Fig 4 of the main article.
(TIFF)

**S6 Fig. Simulation–estimation on the spatial model using fluorescence data only.** (A)–(C) Posterior density in $\alpha$–$\beta$ space for a fit to fluorescence data where the true $P_{\text{CC}} \approx 0.1$, 0.5, 0.9 and the infected cell peak time is held fixed at approximately 18h. We only show densities above a threshold value of $10^{-4}$. (D) Prior density and posterior densities from individual replicates for infected peak time and $P_{\text{CC}}$ with target parameters as specified in (A)–(C). Dashed and solid horizontal lines mark the weighted mean and median values respectively. We also show a box plot of the distribution of posterior weighted means across all four replicates in each case. The replicates in bold are those plotted in (A)–(C). $\alpha$ and $\beta$ have units of $\text{h}^{-1}$ and $(\text{TCID}_{50}/\text{ml})^{-1}\text{h}^{-1}$, respectively.
(TIFF)

**S7 Fig. $\kappa(t)$ for varying diffusion coefficients at fixed values of $P_{\text{CC}}$.** The clustering metric, $\kappa(t)$ for the indicated values of the extracellular viral diffusion coefficient $D$, where and $\alpha$ and $\beta$ are chosen such that $P_{\text{CC}}$ values are approximately 0.1, 0.5, and 0.9 and $t_{\text{peak}}$ is approximately 18h for the specified value of $D$ (according to Table A in S5 Text). We show results from eight simulations in each case. These are the same $\kappa(t)$ trajectories as in Fig 5B–5F in the main text but grouped by $P_{\text{CC}}$. Note that there is some noise associated with the parameter selections for finite diffusion since the lookup tables used are coarser than that for the infinite diffusion model, hence the curves shown only approximately correspond to the indicated $P_{\text{CC}}$ and $t_{\text{peak}}$ values.
(TIFF)

**S8 Fig. Effect of extracellular viral diffusion parameter in observational data on estimates of $\hat{t}_{\text{peak}}$.** Prior density and posterior densities from individual replicates for $t_{\text{peak}}$ for different values of $D$, the value of the extracellular viral diffusion coefficient used in the extended spatial model to generate observational data. We re–fit using the basic spatial model. For each value of $D$ we also show a boxplot of the distribution of posterior weighted means across all four replicates. We show results for the case where the target values of $\alpha$ and $\beta$ give rise to $P_{\text{CC}}$ values of approximately 0.1, 0.5, and 0.9 and $t_{\text{peak}}$ of approximately 18h for the specified value of $D$. $\alpha$ and $\beta$ values for each $D$ values used are specified in Table A in S6 Text. $\alpha$ and $\beta$ have units of $\text{h}^{-1}$ and $(\text{TCID}_{50}/\text{ml})^{-1}\text{h}^{-1}$, respectively.
(TIFF)

**S9 Fig. Effect of sampling size on estimates of $\hat{t}_{\text{peak}}$.** Prior density and posterior densities from individual replicates for $t_{\text{peak}}$ for different values of $S$, the number of cells sampled to calculate the approximation $\kappa_S(t)$ in fitting. For each value of $S$ we also show a boxplot of the distribution of posterior weighted means across all four replicates. We show results for the case where the target values of $\alpha$ and $\beta$ give rise to $P_{\text{CC}}$ values of approximately 0.1, 0.5, and 0.9 and $t_{\text{peak}}$ of approximately 18h. $\alpha$ and $\beta$ have units of $h^{-1}$ and $(\text{TCID}_{50}/\text{ml})^{-1}h^{-1}$, respectively. (TIFF)

**S10 Fig. Posterior predictive check for our parameter estimation for the ODE model, using data from Kongsomros *et al.* [6].** We show the 95% confidence interval of the fluorescent cell trajectories generated from the 8000 posterior samples, along with the specific trajectory of the posterior sample which we have used as our default parameter set throughout the main manuscript. (TIFF)

**S1 Text. ODE model under varying observational noise.** (PDF)

**S2 Text. Spatial model under varying (artificial) observational noise.** (PDF)

**S3 Text. Assigning viral lineage at infection events in the spatial model.** (PDF)

**S4 Text. Simulation–estimation on the spatial model using fluorescence data only.** (PDF)

**S5 Text. Numerical method for the extended spatial model.** (PDF)

**S6 Text. Parameter estimation for the ODE model.** (PDF)

**S7 Text. PMC algorithm for parameter estimation using the spatial model—fluorescence data only.** (PDF)

## Acknowledgments

We are very grateful to Pengxing Cao, Ke Li and Camelia Walker for their insight and guidance in the initial stages of approaching this project, and for valuable discussions about applying Bayesian methods in our work.

## Author Contributions

**Conceptualization:** Thomas Williams, James M. McCaw, James M. Osborne.

**Data curation:** Thomas Williams.

**Formal analysis:** Thomas Williams.

**Funding acquisition:** Thomas Williams, James M. McCaw, James M. Osborne.

**Investigation:** Thomas Williams.

**Methodology:** Thomas Williams, James M. McCaw, James M. Osborne.

**Project administration:** James M. McCaw, James M. Osborne.

**Resources:** James M. McCaw, James M. Osborne.

**Software:** Thomas Williams.

**Supervision:** James M. McCaw, James M. Osborne.

**Validation:** Thomas Williams, James M. McCaw, James M. Osborne.

**Visualization:** Thomas Williams.

**Writing – original draft:** Thomas Williams.

**Writing – review & editing:** Thomas Williams, James M. McCaw, James M. Osborne.

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
