## [Decision Letter · Decision Letter 0]

29 Feb 2024

Dear Dr. Osborne,

Thank you very much for submitting your manuscript "Spatial information allows inference of the prevalence of direct cell-to-cell viral infection" for consideration at PLOS Computational Biology.

As with all papers reviewed by the journal, your manuscript was reviewed by members of the editorial board and by several independent reviewers. In light of the reviews (below this email), we would like to invite the resubmission of a significantly-revised version that takes into account the reviewers' comments.

As you will see, two of the reviewers expressed substantial concerns about the study, and indicated that the approach might not be worked out sufficiently well and that therefore the conclusions are not sufficiently supported by the analysis. One thing I noticed before sending out the paper for review was that some of the literature that aimed to measure the relative contribution of cell free vs cell-to-cell transmission in HIV infection has not been cited, which should be corrected in future versions. This includes:

- Komarova NL, Anghelina D, Voznesensky I, Trinité B, Levy DN, Wodarz D. Relative contribution of free-virus and synaptic transmission to the spread of HIV-1 through target cell populations. Biology letters. 2013 Feb 23;9(1):20121049.

- Kreger J, Garcia J, Zhang H, Komarova NL, Wodarz D, Levy DN. Quantifying the dynamics of viral recombination during free virus and cell-to-cell transmission in HIV-1 infection. Virus evolution. 2021 Jan;7(1):veab026.

We cannot make any decision about publication until we have seen the revised manuscript and your response to the reviewers' comments. Your revised manuscript is also likely to be sent to reviewers for further evaluation.

Sincerely,

Domink Wodarz

Associate Editor

PLOS Computational Biology

Rob De Boer

Section Editor

PLOS Computational Biology

As you will see, two of the reviewers expressed substantial concerns about the study, and indicated that the approach might not be worked out sufficiently well and that therefore the conclusions are not sufficiently supported by the analysis. One thing I noticed before sending out the paper for review was that some of the literature that aimed to measure the relative contribution of cell free vs cell-to-cell transmission in HIV infection has not been cited, which should be corrected in future versions. This includes:

- Komarova NL, Anghelina D, Voznesensky I, Trinité B, Levy DN, Wodarz D. Relative contribution of free-virus and synaptic transmission to the spread of HIV-1 through target cell populations. Biology letters. 2013 Feb 23;9(1):20121049.

- Kreger J, Garcia J, Zhang H, Komarova NL, Wodarz D, Levy DN. Quantifying the dynamics of viral recombination during free virus and cell-to-cell transmission in HIV-1 infection. Virus evolution. 2021 Jan;7(1):veab026.

Reviewer's Responses to Questions

**Comments to the Authors:**

Reviewer #1: The authors present a pair of mathematical models of infection which allow for both free virus and cell-cell transmission, model the measurement noise assuming cellular fluorescence as a determinant of infection, and then numerically simulate the estimation of the cell-cell and cell-virus transmission rates assuming that all other rates are known. The authors show that if spatial information is not gathered (the experiment is assumed to be well-mixed), the resulting parameter estimates have very broad posterior distributions, and include regions where the estimate appears quite confident even when it is incorrect. However, when spatial data is collected (assuming transmission on a 2D culture from a known initial infection point), the posterior distribution of the parameter estimates is constrained and the confidence interval consistently contains the true value.

This is a well-written paper addressing an important topic not just in viral dynamics but also in cell signaling. My only concern is that the authors are only addressing practical identifiability, and they do so without specifically stating this. I would ask them to explicitly state that their conclusions are that alpha and beta are not practically identifiable in the non-spatial system with realistic levels of noise. I would also be interested to know whether the parameters in question are structurally identifiable in equations 1-11, though I wouldn't insist on this for this paper.

Reviewer #2: In this article, the author address the question what type of information might be necessary to reliably infer cell-to-cell transmission dynamics for viral infections. Using a theoretical approach by simulating viral spread with population-based ODE and spatially resolved agent-based models, they show that additional spatial information on mean clustering dynamics of infection can improve the inference of cell-to-cell transmission dynamics in the context of the simultaneous appearance of cell-free and cell-to-cell viral transmission modes.

The study targets a well-defined problem with a clear focus. It is well written and thoroughly structured, although some technical aspects could need some further clarification. The study points towards a useful spatial quantity that could be determined in order to improve inference of cell-to-cell transmission dynamics. However, it is not clear how suitable this approach would be in general, and how applicable it might be given that some aspects influencing these dynamics have not been investigated.

# Major points:

1.) An important aspect which does not seem to be investigated concerns the diffusion rate of the virus. Dependent on fast or slow diffusion, the proposed clustering metric could potentially be impaired in the ability to dissect cell-free and cell-to-cell transmission dynamics. In order to evaluate the general applicability of this single spatial metric in context of viral infections, it might be recommended to investigate how sensitive the inference of the transmission modes would be with regard to the viral diffusion parameter.

2.) The study focuses on the inference of cell-to-cell transmission dynamics within solid epithelial tissue in the context of a target cell limited model. This analysis would neglect cell proliferation and tissue regeneration dynamics, which might influence local cellular densities and, thus, cell interaction dynamics and the growth of infected foci by cell-to-cell transmission. The authors carefully acknowledge the limitations of their approach within the discussion, but the ability to practically infer viral transmission dynamics from undisturbed cultures with a single spatial metric might be impaired given the additional sources of variation. It might be interesting to determine in this simplified model of viral infection dynamics how much observational noise would be actually tolerable and still allowing inference. This could point towards the general applicability of the method with regard to the various aspects (cell heterogeneity, proliferation dynamics, etc) that could impair clear spatial patterns mediated by a given transmission profile.

3.) In this context, the current title of the manuscript also might be a bit too strong/misleading. The authors provide a proof of concept based on simulated data, but an application to actual experimental data would be helpful to show its applicability (or what kind of estimates it would provide in comparison to other approaches cited, that generally predict a high proportion of cell-to-cell transmission in different viral infections, as cited by the authors (e.g. page 1 bottom, page 2 top)).

4.) The parameter inference step, as e.g. shown in Figure 2a-c for the three different parameter combinations and explained within the text, is partly difficult to follow. It is not totally clear how the different estimates (dots) of the different chains were obtained. In addition, I think it would be helpful to show a heat map of the estimates as shown in Figure 4a-c to infer the core area of the distributions. While the inclusion of the proposed clustering metric clearly improves the identification of the targeted quantity P_cc with reduced variation, it is not clear how the identification of the parameters alpha and beta might not be equally affected, given that Figure 2a-c and Figure 4a-c also show the parameters on two different scales. Besides inferring P_cc, it would also be interesting from a quantitative point of view to characterise these two parameters.

# Minor points:

- page 2, second paragraph - There have been additional studies analysing cell-to-cell spread without this strict assumption and in complex environments (e.g. Imle et al. Nat Com 2019).

- page 13/Table 1: As the parameters used for simulation are very precisely indicated in Table 1 (i.e. several digits) and obtained from a fit of a model against experimental data, it seems a a bit irritating to not show the results ("for sake of brevity"). Although one could follow the argument that the actual values are not important, it is not clear if these really indicate realistic values. A comparison to other estimated rates (in this case for influenza infection) should be considered.

Reviewer #3: Williams et al. used mathematical models to identify data types needed for inferring the fraction of cell-to-cell viral infection. The authors first showed that when only fluorescence time series data (representing the level of total infected cells) alone is available, inferring the fraction of cell-to-cell spread is impossible due to the inherent practical (un)identifiability issue with estimating cell-free infection and cell-to-cell infection independently. However, when another type of data, i.e. the proportion of infected neighboring cells over time, is available in addition to the fluorescence time series, inference of the fraction of cell-to-cell spread becomes feasible. The authors further constructed a spatial stochastic model of viral infection and used a Bayesian inference framework to demonstrate the accurate inference of the fraction of cell-to-cell spread using simulated data. Furthermore, the authors conducted sensitivity analysis to show that the inference is reasonably robust even when a small fraction of total cells are measured in the experiment.

The work is well written. Below, I list a few major concerns and some questions that need to be addressed.

1. The authors address an important question: whether a particular parameter of interest (in this case, the fraction of cell-to-cell spread) can be inferred from experimental data? Although the authors mentioned the work is motivated by data collected from Ref. 13, there is no description of the experimental data and the work is on using model inference on hypothetical and simulated datasets. Therefore, a major disconnect is how the framework developed here can be applied to real datasets. Adding more description of the type of datasets that are collected in the lab, discussion of the limitations associated with it (e.g. how noisy a typical data set is, whether time course data is possible for a single experiment, typically what fraction of cells are sampled etc.) are important to put this work into context. I assume that the real dataset is likely different from the simulated datasets in many aspects (e.g. noise level, and the number of time points). Using the framework to perform inference on real datasets (e.g. from Ref. 13) will be also important to demonstrate its utility in dealing with actual data.

2. A key assumption in the model is that the cell-free spread is homogeneous, i.e. every cell is equally likely to be infected by a virus. While I agree this assumption is reasonable for other purposes, it may or may not be a good assumption for the inference here. In essence, the inference is to distinguish non-spatial pattern from spatial pattern of spread. This assumption of homogenous spread of cell-free infection depends on the size of the tissue being sampled and the diffusion of the virus relative to the size of the tissue. When the diffusion of the virus is low (in certain medium or in vivo), one will expect cell-free infection leads to infection of cells close by (as well as neighboring cells). This has been suggested for influenza infection in the upper respiratory tract and HCV infection in the liver. In this case, cell-free infection may look like cell-to-cell infection. I think it is important to discuss this complication and that the inference is not inferring cell-to-cell infection per se; rather it’s an inference of spatial vs. non-spatial spread. Therefore, the inference may represent an upper bound for the fraction of true cell-to-cell spread. Also, I think this consideration may become important especially when only a fraction of cells are sampled in the experiment – in this case, one likely samples cells close by (rather than neighboring cells). Can the authors extend the spatial model to test how sensitive the inference to the homogenous assumption by assuming cell-to-cell spread follows diffusion pattern (where the diffusion constant is in a biologically plausible range), especially when a fraction of cells are sampled?

3. It is not clear how the clustering metric k_i(t) is calculated in the scenario where only a fraction of cells are sampled. I assume the fraction is calculated by counting the sampled cells only in both numerator and denominator – this is not written in the manuscript. In this case, what qualifies a neighboring cell? I imagine when the fraction of sampled cells is low, none (or only few) of the sampled cells are ‘actually’ neighboring to each other? I think it is also important to discuss why the inference is still robust when only 50 or 100 cells out of 2500 cells are sampled (for whatever the definition of neighboring cell it is). Again, this question also points to the importance of testing the robustness of the inference when cell-free infection is not homogenous (as assumed).

4. (A minor point) I believe the framework presented here only applies to viral infections where the target cells do not move. It will not work for infections such as HIV (where target cells move constantly). I think the scope of the framework should be mentioned throughout the manuscript.

**Have the authors made all data and (if applicable) computational code underlying the findings in their manuscript fully available?**

The PLOS Data policy requ

---

## [Decision Letter · Decision Letter 1]

19 Jun 2024

Dear Dr. Osborne,

We are pleased to inform you that your manuscript 'Spatial information allows inference of the prevalence of direct cell-to-cell viral infection' has been provisionally accepted for publication in PLOS Computational Biology.

Best regards,

Dominik Wodarz

Academic Editor

PLOS Computational Biology

Rob De Boer

Section Editor

PLOS Computational Biology

Reviewer's Responses to Questions

**Comments to the Authors:**

Reviewer #3: I do not have further comments/concerns.

**Have the authors made all data and (if applicable) computational code underlying the findings in their manuscript fully available?**

Reviewer #3: None

PLOS authors have the option to publish the peer review history of their article (what does this mean?). If published, this will include your full peer review and any attached files.

Reviewer #3: **Yes: **Ruian Ke

---

## [Editor Report · Acceptance letter]

17 Jul 2024

PCOMPBIOL-D-23-01754R1 

Spatial information allows inference of the prevalence of direct cell-to-cell viral infection

Dear Dr Osborne,

I am pleased to inform you that your manuscript has been formally accepted for publication in PLOS Computational Biology. Your manuscript is now with our production department and you will be notified of the publication date in due course.

With kind regards,

Anita Estes
